# Distributionally Robust Optimization with Data Geometry

**Jiashuo Liu**[1,*]**, Jiayun Wu**[1,*]**, Bo Li**[2]**, Peng Cui**[1,†]

[1] Department of Computer Science & Technology, Tsinghua University, Beijing, China
[2]School of Economics and Management, Tsinghua University, Beijing, China
liujiashuo77@gmail.com,jiayun.wu.work@gmail.com
libo@sem.tsinghua.edu.cn, cuip@tsinghua.edu.cn

## Abstract

Distributionally Robust Optimization (DRO) serves as a robust alternative to empirical risk minimization (ERM), which optimizes the worst-case distribution in an uncertainty set typically specified by distance metrics including $f$-divergence and the Wasserstein distance. The metrics defined in the ostensible high dimensional space lead to exceedingly large uncertainty sets, resulting in the underperformance of most existing DRO methods. It has been well documented that high dimensional data approximately resides on low dimensional manifolds. In this work, to further constrain the uncertainty set, we incorporate data geometric properties into the design of distance metrics, obtaining our novel Geometric Wasserstein DRO (GDRO). Empowered by Gradient Flow, we derive a generically applicable approximate algorithm for the optimization of GDRO, and provide the bounded error rate of the approximation as well as the convergence rate of our algorithm. We also theoretically characterize the edge cases where certain existing DRO methods are the degeneracy of GDRO. Extensive experiments justify the superiority of our GDRO to existing DRO methods in multiple settings with strong distributional shifts, and confirm that the uncertainty set of GDRO adapts to data geometry.

## 1 Introduction

Machine learning algorithms with empirical risk minimization often suffer from poor generalization performance under distributional shifts in real applications due to the widespread latent heterogeneity, domain shifts, and data selection bias, *etc*. It is demanded for machine learning algorithms to achieve uniformly good performances against potential distributional shifts, especially in high-stake applications. Towards this goal, distributionally robust optimization (DRO) [27, 23, 29, 4, 13, 11], stemming from the literature of robust learning, has been proposed and developed in recent years. It optimizes the worst-case distribution within an uncertainty set $\mathcal{P}(P_{tr})$ lying around the training distribution $P_{tr}$. When the testing distribution $P_{te}$ is contained in $\mathcal{P}(P_{tr})$, DRO could guarantee the generalization performance on $P_{te}$.

In principle, the effectiveness of DRO heavily depends on the rationality of its uncertainty set $\mathcal{P}(P_{tr})$ which is commonly formulated as a ball surrounding the training distribution endowed with a certain distance metric. An ideal uncertainty set should be constituted by all realistic distributions that may be encountered in test environments. However, existing DRO methods adopting the Wasserstein distance (i.e. WDRO methods [27, 29, 4, 13]) or $f$-divergence distance (i.e. $f$-DRO methods [23, 11]) tend to generate *over-flexible* uncertainty sets that incorporate unrealistic distributions far beyond the ideal

---

*Equal Contributions
†Corresponding Author

36th Conference on Neural Information Processing Systems (NeurIPS 2022).

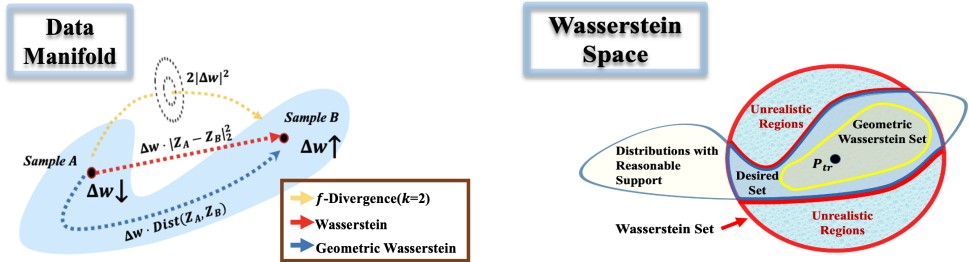

(a) Comparison of metrics on data manifold.  (b) Comparison of uncertainty sets.

Figure 1: Toy illustrations. Figure (a) illustrates the transportation paths and corresponding costs of the distance metrics. (b) depicts the uncertainty set of WDRO and GDRO compared with an ideal uncertainty set in Wasserstein space, where each point denotes a distribution.

uncertainty set [15, 13]. As such unrealistic distributions must violate the underlying predicting mechanism, they are prone to be the worst-case and attract much optimization energy in the DRO framework, making the learned model deviate from the true predicting mechanism.

Here we argue that the unrealistic distributions mentioned above originate from the distance metrics' inherent ignorance of data geometry, as illustrated in Figure 1. The Euclidean-norm transportation cost measured by the $L_2$-Wasserstein metric leads to a straight-line transportation path as shown in Figure 1(a) (red dotted line) which deviates from the data manifold (blue region). Therefore, WDRO methods tend to create unrealistic samples beyond the underlying data manifold, resulting in unrealistic distributions. $f$-divergence can also be interpreted as a data geometry-independent measure of the transportation cost confined in the support of $P_{tr}$. Taking $\chi^2$-divergence for example, the cost is a constant to transfer per unit of probability weights between samples, like a virtual tunnel (yellow dotted line in Figure 1(a)). In such a case, the noisy samples (e.g. outliers or samples with label noises) are more prone to be the worst-case and thus gather much larger weights than normal samples. The resultant distribution is obviously unrealistic.

To mitigate the problem, it is imperative to introduce a new distance metric incorporating data geometry to further constrain the uncertainty set and avoid the undesired cases. As illustrated in Figure 1(a), considering the common assumption that data lie on a low-dimensional manifold [24, 30, 2], we expect the probability density transportation path (the blue dotted line) is *restricted within the data manifold* (the blue region). In this way, the uncertainty set (i.e. the Geometric Wasserstein Set as shown in Figure 1(b)) could inherently exclude the distributions beyond the data manifold. Furthermore, it is harder to gather probability weights on isolated noisy samples, which also mitigate the undesired cases in $f$-DRO.

In this work, we propose a novel Geometric Wasserstein DRO (GDRO) method by exploiting the discrete Geometric Wasserstein distance [6] which measures the transportation cost of probability density along the geodesic in a metric space. As the Geometric Wasserstein distance does not enjoy an analytical expression, we derive an approximate algorithm from the Gradient Flow in the Finsler manifold endowed with Geometric Wasserstein Distance (in section 3.2). We further theoretically specify an exponentially vanishing error rate of our approximation as well as a $O(1/\sqrt{T})$ convergence rate of our algorithm, and characterize the edge cases where GDRO will degenerate to $f$-DRO or Wasserstein DRO (in section 3.3 and 3.4). Comprehensive experiments encompassing various distributional shifts, including sub-population shifts and class difficulty shifts, validate the effectiveness of our proposed GDRO (in section 4). We also observe a lower Dirichlet Energy (i.e. higher smoothness) of GDRO's estimated worst-case distribution w.r.t the data manifold compared with existing DRO methods, justifying its adaptability to data geometry.

## 2 Preliminaries on Distributionally Robust Optimization

**Notations.** $X \in \mathcal{X}$ denotes the covariates, $Y \in \mathcal{Y}$ denotes the target, $f_\theta(\cdot) : \mathcal{X} \to \mathcal{Y}$ is the predictor parameterized by $\theta \in \Theta$. $P_{tr}(X, Y)$ and $P_{te}(X, Y)$ abbreviated with $P_{tr}$ and $P_{te}$ respectively represent the joint training distribution and test distribution. The random variable of data points is denoted by $Z = (X, Y) \in \mathcal{Z}$.

Distributionally Robust Optimization (DRO) is formulated as:

$$\theta^* = \arg\min_{\theta \in \Theta} \sup_{P \in \mathcal{P}(P_{tr})} \mathbb{E}_P[\ell(f_\theta(X), Y)], \tag{1}$$

where $\ell$ is a loss function, $\mathcal{P}(P_{tr}) = \{P : Dist(P, P_{tr}) \leq \epsilon\}$ characterizes the uncertainty set surrounding the training distribution restricted by a radius $\epsilon$, and *Dist* is a distance metric between probability distributions. Most works specify the *Dist* metric as the $f$-divergence [23, 11] or the Wasserstein distance [27, 29, 4, 26, 12].

$f$**-divergence DRO** (*abbr.* $f$-DRO) $\quad f$-divergence is defined as $D_f(P\|Q) = \int f(dP/dQ)dQ$, where $f(\cdot)$ is a convex function and $f(1) = 0$. Two typical instances of $f$-divergences are KL-divergence ($f(t) = t \log t$) and $\chi^2$-divergence ($f(t) = (t-1)^2$). [23] theoretically demonstrates the equivalence between $\chi^2$-DRO and the variance-regularized empirical risk minimization (ERM) problem, and [11] derives the optimization algorithm for a family of $f$-DRO. However, as proven in [15], $f$-DRO faces the over-pessimism problem and ends up giving a classifier only fitting the given training distribution, which we attribute to the ignorance of data geometry. As shown in Figure 1(a), $f$-divergence only cares about the probability of each sample (only $dP, dQ$ occur). However, data geometry information is crucial for a reasonable uncertainty set, since it is well-accepted that data lie on a low-dimensional manifold and adjacent data points have similar degrees of importance. For example, for heterogeneous data, while one hopes to focus on some sub-population (e.g., put more weights on a group of data), without data geometric information, the distribution in the $f$-divergence ball is prone to only focus on some isolated samples with higher noises (as shown in Figure 3(a)). And in Figure 3(b), we find the worst-case distribution of $f$-DRO (with KL-divergence) is not smooth (with larger Dirichlet Energy) w.r.t. the data manifold.

**Wasserstein DRO** (*abbr.* WDRO) $\quad$ Compared with the $f$-divergence ball that does not extend the support of the training distribution, the uncertainty set built with Wasserstein distance allows for the extension of the support [27, 29, 4]. [26, 12, 4] convert the original problem into a regularized ERM problem, but it is suitable only for a limited class of loss functions and transportation cost functions. [29] proposes an approximate optimization method for Wasserstein DRO that could be applied to deep neural networks, which protects models from adversarial attacks. However, the flexibility of the Wasserstein ball also causes an over-pessimistic estimation under strong distributional shifts [13], where the created samples are too noisy to obtain a confident model. As demonstrated in Figure 3(a), WDRO adds much more noises to the data and thus hurts the generalization performances in practice.

Therefore, to mitigate the over-pessimism problem of DRO, we propose to incorporate the geometric properties into the uncertainty set. Compared with traditional shape-constrained methods [17, 16] for multivariate extreme event analysis that use the unimodality to constitute the uncertainty set, our proposed method characterizes the data manifold in a data-driven way and incorporates it into the DRO framework intrinsically via the Geometric Wasserstein distance metric, which is also compatible with manifold learning and graph learning methods.

## 3 Proposed Method

In this work, we propose Distributionally Robust Optimization with Geometric Wasserstein distance (GDRO). In the following of this section, we first introduce the Geometric Wasserstein distance and propose the overall objective of GDRO; then we derive an approximate algorithm for optimization utilizing Gradient Flow; finally, some theoretical properties are proved and connections with existing DRO methods are demonstrated.

### 3.1 Discrete Geometric Wasserstein Distance $\mathcal{GW}_{G_0}$ and GDRO

We firstly introduce the Discrete Geometric Wasserstein distance, which extends the Benamou-Brenier formulation of the optimal transport problem to a metric space. The first step is to define a discrete velocity field and its discrete divergence, which we mainly follow the construction by Chow *et al.* [6].

Consider a given weighted finite graph $G_0 = (V, E, w)$ with $n$ nodes, where $V = \{1, 2, \ldots, n\}$ is the vertex set, $E$ is the edge set and $w = (w_{ij})_{i,j \in V}$ is the weight of each edge. A *velocity field* $v = (v_{ij})_{i,j \in V} \in \mathbb{R}^{n \times n}$ on $G_0$ is defined to be a skew-symmetric matrix on the edge set $E$ such that $v_{ij} = -v_{ji}$ if $(i, j) \in E$. The probability set (simplex) $\mathscr{P}(G_0)$ supported on $V$ is defined as $\mathscr{P}(G_0) = \{(p_i)_{i=1}^n \in \mathbb{R}^n | \sum_{i=1}^n p_i = 1, p_i \geq 0, \text{for any } i \in V\}$ and its interior

is denoted by $\mathscr{P}_o(G_0)$. $\kappa_{ij}$ is a predefined "cross-sectional area" typically interpolated with the associated nodes' densities $p_i, p_j$. The direct approach is to take the arithmetic average such that $\kappa_{ij}(p) = (p_i + p_j)/2$. However, to ensure the positiveness of $p$ during optimization, we adopt the *upwind interpolation*:$\kappa_{ij}(p) = \mathbb{I}(v_{ij} > 0)p_j + \mathbb{I}(v_{ij} \leq 0)p_i$. One could thereafter define the product $pv \in \mathbb{R}^{n \times n}$, called *flux function* on $G_0$, by $pv := (v_{ij}\kappa_{ij}(p))_{(i,j) \in E}$. The *divergence* of $pv$ is $\mathrm{div}_{G_0}(pv) := -(\sum_{j \in V:(i,j) \in E} \sqrt{w_{ij}}v_{ij}\kappa_{ij}(p))_{i=1}^n$ which is a vector in $\mathbb{R}^n$. The divergence vector is supposed to lie in the tangent space of $\mathscr{P}_o(G_0)$, summing over all the in-fluxes and out-fluxes along edges of a certain node, with each edge transporting a probability density $\sqrt{w_{ij}}v_{ij}\kappa_{ij}(p)$. Now we are ready to define Geometric Wasserstein distance in Equation 2.

**Definition 3.1** (Discrete Geometric Wasserstein Distance $\mathcal{GW}_{G_0}(\cdot, \cdot)$ [6]). *Given a finite graph $G_0$, for any pair of distributions $p^0, p^1 \in \mathscr{P}_o(G_0)$, define the Geometric Wasserstein Distance:*

$$\mathcal{GW}_{G_0}^2(p^0, p^1) := \inf_v \left\{ \int_0^1 \frac{1}{2} \sum_{(i,j) \in E} \kappa_{ij}(p)v_{ij}^2 dt : \frac{dp}{dt} + div_{G_0}(pv) = 0, p(0) = p^0, p(1) = p^1 \right\}, \quad (2)$$

*where $v \in \mathbb{R}^{n \times n}$ denotes the velocity field on $G_0$, $p$ is a continuously differentiable curve $p(t) : [0,1] \to \mathscr{P}_o(G_0)$, and $\kappa_{ij}(p)$ is a pre-defined interpolation function between $p_i$ and $p_j$.*

Intuitively $v$ is a velocity field continuously transporting masses to convert the density distribution from $p^0$ to $p^1$ along a curve in the Wasserstein space [31]. Equation 2 measures the shortest (geodesic) length among all possible plans, which is calculated by integrating a total "kinetic energy" of the velocity field over the transportation process. Compared with the Benamou-Brenier formulation of continuous $L_2$-Wasserstein distance, it ensures that the transportation path *stays within the manifold* (as the blue dotted line shown in Figure 1(a)), and it induces a smoother estimate of the worst-case probability distribution w.r.t the data structure since weights are exchanged just between neighbors.

Then we present the *overall objective function* of Distributionally Robust Optimization with Geometric Wasserstein distance (GDRO). Given the training dataset $D_{tr} = \{(x_i, y_i)\}_{i=1}^n$ and its empirical marginal distribution $\hat{P}_{tr} = \frac{1}{n} \sum_i \delta(x_i)$, along with a manifold structure represented by graph $G_0$, we intend to obtain a distributionally robust predictor parameterized by $\theta^*$ such that for certain $\epsilon > 0$:

$$\theta^* = \arg\min_{\theta \in \Theta} \sup_{P:\mathcal{GW}_{G_0}^2(\hat{P}_{tr}, P) \leq \epsilon} \left\{ \mathcal{R}_n(\theta, p) = \sum_{i=1}^n p_i \ell(f_\theta(x_i), y_i) - \beta \sum_{i=1}^n p_i \log p_i \right\}. \quad (3)$$

We add a minor entropy-regularization with a small $\beta$ as proposed in the entropy-balancing literature [14] to avoid singular cases and ensure the convergence of our optimization in section 3.2. Owing to the Geometric Wasserstein distance, the uncertainty set of GDRO excludes those distributions supported on points beyond the data manifold and the Geometric Wassserstein Ball is directional in Wasserstein space as it stretches along the data structure, as depicted in Figure 1(b).

**How is $G_0$ estimated?** To characterize the data manifold, the $G_0$ used in GDRO is constructed as a k-nearest neighbor (kNN) graph *from the training data only*, as the kNN graph is shown to have a good approximation of the geodesic distance within local structures on the manifold [21, 7]. Note that our GDRO is *compatible with any manifold learning and graph learning methods*.

### 3.2 Optimization

In this subsection, we derive the optimization algorithm for GDRO. Due to the lack of an analytical form of the Geometric Wasserstein distance, we give up providing a prescribed amount $\epsilon$ of robustness

---

**Algorithm 1** Geometric Wasserstein Distributionally Robust Optimization (GDRO)

---

**Input:** Training Dataset $D_{tr} = \{(x_i, y_i)\}_{i=1}^n$, learning rate $\alpha_\theta$, gradient flow iterations $T$, entropy term $\beta$, manifold representation $G_0$ (learned by kNN algorithm from $D_{tr}$).

**Initialization:** Sample weights initialized as $(1/n, \ldots, 1/n)^T$. Predictor's parameters initialized as $\theta^{(0)}$.

**for** $i = 0$ **to** Epochs **do**
    1. Simulate gradient flow for $T$ time steps according to Equation 5~6 to learn an approximate worst-case probability weight $p^T$.
    2. $\theta^{(i+1)} \leftarrow \theta^{(i)} - \alpha_\theta \nabla_\theta (\sum_i p_i^T \ell_i(\theta))$
**end for**

---

in Equation 3 and propose an alternate optimization algorithm as an approximation. For fixed probability weights $p$, the parameter $\theta$ could be optimized via gradient descents for $\mathcal{R}_n(\theta, p)$ w.r.t. $\theta$ in parameter space $\Theta$. The inner supremum problem can be approximately solved via gradient ascents for $\mathcal{R}_n(\theta, p)$ w.r.t. $p$ in the *Geometric Wasserstein space* $(\mathscr{P}_o(G_0), \mathcal{GW}_{G_0})$. And the cost measured by $\mathcal{GW}^2_{G_0}(\hat{P}_{tr}, \cdot)$ could be approximated with the length of the gradient flow, which is a curve in $(\mathscr{P}_o(G_0), \mathcal{GW}_{G_0})$.

Here we clarify some notations. $p : [0, T] \mapsto \mathscr{P}_o(G_0)$ denotes the *continuous gradient flow*, and the probability weight of the $i$-th sample at time $t$ is abbreviated as $p_i(t)$. The *time-discretized gradient flow* corresponding with the time step $\tau$ is denoted as $\hat{p}_\tau : [0, T] \mapsto \mathscr{P}_o(G_0)$, and $\hat{p}_\tau(t)$ is abbreviated as $\hat{p}_\tau^t$. For the optimization, we adopt the time-discretized definition of Gradient Flow [31] for $-\mathcal{R}_n(\theta, p)$ in the Geometric Wasserstein space $(\mathscr{P}_o(G_0), \mathcal{GW}_{G_0})$ as: (with the time step $\tau$)

$$\hat{p}_\tau(t + \tau) = \arg \max_{p \in \mathscr{P}_o(G_0)} \mathcal{R}_n(\theta, p) - \frac{1}{2\tau}\mathcal{GW}^2_{G_0}(\hat{p}_\tau(t), p). \tag{4}$$

When $\tau \to 0$, the time-discretized gradient flow $\hat{p}_\tau$ becomes the continuous one $p$. Note that Equation 4 describes the Gradient Flow as a steepest ascent curve locally optimizing for a maximal objective within an infinitesimal Geometric Wasserstein ball, and it coincides with the Lagrangian penalty problem of Equation 3. In theorem 3.1 we would prove that Equation 4 finds the exact solution to a *local* GDRO at each time step.

Following Chow *et al.* [6], the analytical solution to Equation 4 as $\tau \to 0$ could be derived as:

$$\frac{dp_i}{dt} = \sum_{j:(i,j)\in E} w_{ij}\kappa_{ij}(\ell_i - \ell_j) + \beta \sum_{j:(i,j)\in E} w_{ij}\kappa_{ij}(\log p_j - \log p_i), \tag{5}$$

where $p_i$ denotes the time-dependent probability function of the $i$-th sample, $l_i$ denotes the loss of the $i$-th sample and we take an upwind interpolation of $\kappa$: $\kappa_{ij}(p) = \mathbb{I}(v_{ij} > 0)p_j + \mathbb{I}(v_{ij} \leq 0)p_i$, so that the probability density transferred on an edge equals the density from the origin node associated with the velocity field. The upwind interpolation guarantees that the probability weight $p$ stays positive along the Gradient Flow in Equation 5. Then we discretize equation 5 with Forward Euler Method:

$$p_i(t + \alpha) = p_i(t) + \alpha dp_i(t)/dt, \tag{6}$$

where $\alpha$ is a learning rate. For our algorithm, we control the maximum time step as $t \leq T$ in Equation 6 to approximately restrict the radius of the Geometric Wasserstein ball. We prove in theorem 3.2 that for the final time step $t = T$, the probability weights $p(T)$ learned by Equation 6 guarantees a *global* error rate $e^{-CT}$ from the worst-case risk $\mathcal{R}_n(\theta, p^*)$ constrained in an $\epsilon(\theta)$-radius ball where $\epsilon(\theta) = \mathcal{GW}^2_{G_0}(\hat{P}_{tr}, p(T))$ and $p^* = \arg \sup_p\{\mathcal{R}_n(\theta, p) : \mathcal{GW}^2_{G_0}(\hat{P}_{tr}, p) \leq \epsilon(\theta)\}$. The result is similar to conventions in WDRO [29], which gives up providing a prescribed radius of its uncertainty set but turns to an approximation with a intermediate hyperparameter. Pseudo-code of the whole algorithm is shown in Algorithm 1. The whole derivations are in Appendix.

### 3.3 Theoretical Properties

In this section, we prove the equivalence between our Gradient-Flow-based algorithm and a local GDRO problem, and the bound of its global error rate as well as the convergence rate is derived. We first provide the robustness guarantee for the Lagrangian penalty problem in Equation 4.

**Theorem 3.1** (Local Robustness Guarantees of Lagrangian Penalty Problem). *For any $\tau > 0, t > 0$ and given $\theta$, denote the solution of Equation 4 as $p^*(\theta) = \arg \sup_{p \in \mathscr{P}(G_0)} \mathcal{R}_n(\theta, p) - \frac{1}{2\tau}\mathcal{GW}^2_{G_0}(\hat{p}_\tau^t(\theta), p)$. Let $\epsilon_\tau(\theta) = \mathcal{GW}^2_{G_0}(\hat{p}_\tau^t(\theta), p^*(\theta))$, we have*

$$\sup_{p \in \mathscr{P}_o(G_0)} \mathcal{R}_n(\theta, p) - \frac{1}{2\tau}\mathcal{GW}^2_{G_0}(\hat{p}_\tau^t(\theta), p) = \sup_{p:\mathcal{GW}^2_{G_0}(\hat{p}_\tau^t(\theta), p) \leq \epsilon_\tau(\theta)} \mathcal{R}_n(\theta, p). \tag{7}$$

Theorem 3.1 proves that at each time step our Lagrangian penalty problem is equivalent to a local GDRO within the $\epsilon_\tau(\theta)$-radius Geometric Wasserstein ball. It further shows that with $\tau \to 0$ in Equation 4, our gradient flow constantly finds the steepest descent direction. Then we theoretically analyze the global error rate brought by our approximate algorithm.

**Theorem 3.2** (Global Error Rate Bound). *Given the model parameter $\theta$, denote the approximate worst-case by gradient descent in Equation 6 after time $t$ as $p^t(\theta)$, and $\epsilon(\theta) = \mathcal{GW}^2_{G_0}(\hat{P}_{tr}, p^t(\theta))$ denotes the distance between our approximation $p^t$ and the training distribution $\hat{P}_{tr}$. Then denote*

*the real worst-case distribution within the $\epsilon(\theta)$-radius discrete Geometric Wasserstein-ball as $p^*(\theta)$, that is,*

$$p^*(\theta) = \arg \sup_{p:\mathcal{GW}^2_{G_0}(\hat{P}_{tr},p)\leq\epsilon(\theta)} \sum_{i=1}^{n} p_i\ell_i - \beta \sum_{i=1}^{n} p_i \log p_i. \tag{8}$$

*Here we derive the bound w.r.t. the error ratio of objective function $R_n(\theta,p)$ (abbr. $\mathcal{R}(p)$). For $\theta \in \Theta$, there exists $C > 0$ such that*

$$\text{Error Rate} = \left(\mathcal{R}(p^*) - \mathcal{R}(p^t)\right) / \left(\mathcal{R}(p^*) - \mathcal{R}(\hat{P}_{tr})\right) < e^{-Ct}, \tag{9}$$

*and when $t \to \infty$, Error Rate $\to 0$. The value of $C$ depends on $\ell, \beta, n$.*

Theorem 3.2 theoretically characterizes 'how far' our approximation $p^t$ is from the real worst-case $p^*$ in terms of the drop ratio of the objective function $\mathcal{R}(p)$. At last we derive the convergence rate of our Algorithm 1.

**Theorem 3.3** (Convergence of Algorithm 1)**.** *Denote the objective function for the predictor as:*

$$F(\theta) = \sup_{\mathcal{GW}^2_{G_0}(\hat{P}_{tr},p)\leq\epsilon(\theta)} \mathcal{R}_n(\theta,p), \tag{10}$$

*which is assumed as $L$-smooth and $\mathcal{R}_n(\theta,p)$ satisfies $L_p$-smoothness such that $\|\nabla_p \mathcal{R}_n(\theta,p) - \nabla_p \mathcal{R}_n(\theta,p')\|_2 \leq L_p\|p - p'\|_2$. $\epsilon(\theta)$ follows the definition in Theorem 3.2. Take a constant $\Delta_F \geq F(\theta^{(0)}) - \inf_\theta F(\theta)$ and set step size as $\alpha = \sqrt{\Delta_F/(LK)}$. For $t \geq T_0$ where $T_0$ is a constant, denote the upper bound of $\|p^t - p^*\|_2^2$ as $\gamma$ and train the model for $K$ steps, we have:*

$$\frac{1}{K}\mathbb{E}\left[\sum_{k=1}^{K} \|\nabla_\theta F(\theta^{(k)})\|_2^2\right] - \frac{(1+2\sqrt{L\Delta_F/K})}{1-2\sqrt{L\Delta_F/K}} L_p^2 \gamma \leq \frac{2\Delta_F}{\sqrt{\Delta_F K} - 2L\Delta_F}. \tag{11}$$

Here we make a common assumption on the smoothness of the objective function as in [29]. As $K \to \infty$, $\nabla_\theta F(\theta^{(k)})$ will achieve a square-root convergence only if $\gamma$ is controlled by the exponentially vanishing error rate in Theorem 3.2. And the accuracy parameter $\gamma$ remains a fixed effect on optimization accuracy.

### 3.4 Connections with Conventional DRO Methods

In Theorem 3.4, we illustrate the connections of our GDRO with $f$-DRO.

**Theorem 3.4** (Connection with $f$-DRO with KL-diveregence (KL-DRO).)**.** *Relax the discrete Geometric Wasserstein-ball regularization (set $\epsilon \to \infty$) and set the graph $G_0$ to a fully-connected graph, and then the solution of GDRO is equivalent to the following form of KL-DRO:*

$$\min_{\theta\in\Theta} \sup_{p:D_{KL}(p\|\hat{P}_{tr})\leq\hat{\epsilon}(\theta)} \sum_{i=1}^{n} p_i\ell(f_\theta(x_i),y_i), \quad \text{with } \hat{\epsilon}(\theta) = D_{KL}(p^*(\theta)\|\hat{P}_{tr}), \tag{12}$$

*where $p^*(\theta) = \arg\max_p \sum_{i=1}^{n} p_i\ell(f_\theta(x_i),y_i) - \beta\sum_{i=1}^{n} p_i \log p_i$.*

**Remark** (Connections with WDRO)**.** *Since conventional WDRO allows distributions to extend training support, our proposed GDRO is intrinsically different from WDRO. Intuitively, for infinite samples, if the graph $G_0$ is set to a fully-connected graph with edge weights $w_{ij} = \|z_i - z_j\|^2$ and $\beta$ is set to 0, our GDRO resembles support-restricted version of WDRO.*

## 4 Experiments

In this section, we investigate the empirical performance of our proposed GDRO on different simulation and real-world datasets under various kinds of distributional shifts, including *sub-population shifts* and *class difficulty shifts*. As for baselines, we compare with empirical risk minimization (ERM), WDRO [4, 29] and two typical $f$-DRO methods [11], including KL-DRO ($f(t) = t \ln t$) and $\chi^2$-DRO ($f(t) = (t-1)^2$).

**Implementation Details** For all experiments, $G_0$ is constructed as a k-nearest neighbor graph from the *training data only* at the initialization step. Specifically, we adopt NN-Descent [10] to efficiently estimate the k-nearest neighbor graph for the large-scale dataset Colored MNIST while performing

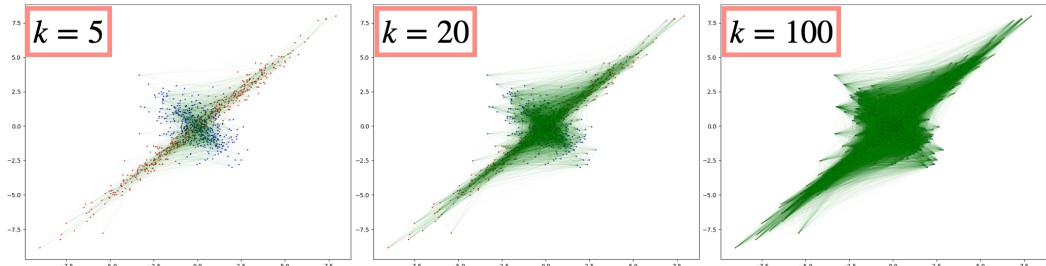

Figure 2: Visualization of learned kNN graph with different $k$ of the regression data, which is projected on the plane spanned by the unit vector of $V$ axis and $\theta_S$ with a projection matrix $\begin{bmatrix} \mathbf{0_{1,5}} & \mathbf{0_{1,4}} & 1 \\ \theta_S^T & \mathbf{0_{1,4}} & 0 \end{bmatrix}$.

an exact search for k-nearest neighbors in the other experiments. We adopt MSE as the empirical loss function for regression tasks and cross-entropy for classification tasks. We use MLPs for the Colored MNIST and Ionosphere datasets, and linear models in the other experiments. Besides, we find that the two-stage optimization is enough for good performances, as mentioned in [19], and we use it in our experiments. Note that GDRO is *compatible with any parameterized models including deep models*. The simulation of gradient flow in Equation 6 is implemented by message propagation with DGL package [32], which scales linearly with sample size and enjoys parallelization by GPU.

## 4.1 Simulation Data

In this subsection, we use simulations to verify that our GDRO could deal with sub-population shifts and to some extent resist the label noises. And we also visualize the effects of the kNN algorithm as well as the sensitivity of GDRO to the parameter $k$.

Table 1: Results on the Selection Bias Experiments. We report the root mean square errors.

| | Train(major) | Train(minor) | Test | | | | | | | Parameter |
|---|---|---|---|---|---|---|---|---|---|---|
| **Simulation 1**: regression data without label noises | | | | | | | | | | |
| Bias Ratio $r$ | $r = 1.9$ | $r = -1.3$ | $r = -1.5$ | $r = -1.7$ | $r = -1.9$ | $r = -2.3$ | $r = -2.7$ | $r = -3.0$ | | Est Error |
| ERM | **0.339** | 0.876 | 0.892 | 0.884 | 0.864 | 0.880 | 0.843 | 0.888 | | 0.423 |
| WDRO | **0.339** | 0.877 | 0.894 | 0.885 | 0.865 | 0.882 | 0.844 | 0.890 | | 0.424 |
| $\chi^2$-DRO | 0.411 | 0.744 | 0.757 | 0.741 | 0.733 | 0.742 | 0.714 | 0.755 | | 0.367 |
| KL-DRO | 0.370 | 0.713 | 0.728 | 0.716 | 0.708 | 0.713 | 0.685 | 0.724 | | 0.319 |
| GDRO | 0.493 | **0.492** | **0.508** | **0.489** | **0.501** | **0.483** | **0.486** | **0.496** | | **0.033** |
| **Simulation 2**: regression data with label noises | | | | | | | | | | |
| ERM | **0.335** | 0.845 | 0.885 | 0.879 | 0.874 | 0.884 | 0.882 | 0.876 | | 0.422 |
| WDRO | **0.335** | 0.896 | 0.887 | 0.880 | 0.875 | 0.886 | 0.884 | 0.877 | | 0.423 |
| $\chi^2$-DRO | 0.375 | 0.866 | 0.855 | 0.856 | 0.843 | 0.860 | 0.854 | 0.845 | | 0.408 |
| KL-DRO | 0.393 | 0.879 | 0.868 | 0.866 | 0.856 | 0.876 | 0.866 | 0.861 | | 0.391 |
| GDRO | 0.542 | **0.537** | **0.553** | **0.549** | **0.534** | **0.539** | **0.555** | **0.550** | | **0.058** |
| **Simulation 3**: vary $k$ under Simulation 1 | | | | | | | | | | |
| GDRO ($k = 5$) | 0.493 | 0.492 | 0.508 | 0.489 | 0.501 | 0.483 | 0.486 | 0.496 | | 0.033 |
| GDRO ($k = 20$) | 0.518 | 0.507 | 0.521 | 0.502 | 0.514 | 0.497 | 0.504 | 0.508 | | 0.036 |
| GDRO ($k = 100$) | 0.379 | 0.673 | 0.688 | 0.676 | 0.670 | 0.672 | 0.647 | 0.683 | | 0.286 |

## 1. Regression: Sub-population Shifts via Selection Bias Mechanism

**Data Generation** The input features $X = [S, U, V]^T \in \mathbb{R}^{10}$ are comprised of stable features $S \in \mathbb{R}^5$, noisy features $U \in \mathbb{R}^4$ and the spurious feature $V \in \mathbb{R}$:

$$S \sim \mathcal{N}(0, 2\mathbb{I}_5) \in \mathbb{R}^5, \quad U \sim \mathcal{N}(0, 2\mathbb{I}_4) \in \mathbb{R}^4, \quad Y = \theta_S^T S + 0.1 \cdot S_1 S_2 S_3 + \mathcal{N}(0, 0.5), \quad (13)$$

$$V \sim \text{Laplace}(\text{sign}(r) \cdot Y, \ \frac{1}{5 \ln |r|}) \in \mathbb{R}, \quad (14)$$

where $\theta_S \in \mathbb{R}^5$ is the coefficient of the true model. $|r| > 1$ is a factor for each sub-population. $S$ are *stable features* with the invariant relationship with $Y$. $U$ are *noisy features* such that $U \perp Y$. And $V$ is the *spurious feature* whose relationship with $Y$ is unstable and is controlled by the factor $r$. Intuitively, $\text{sign}(r)$ controls whether the spurious correlation between $V$ and $Y$ is positive or negative. And $|r|$ controls the strength of the spurious correlation: the larger $|r|$ is, the stronger the spurious correlation is.

**Simulation Setting 1**   In training, we generate 10000 points, where the major group contains 95% data with $r = 1.9$ (i.e. strong positive spurious correlation) and the minor group contains 5% data with $r = -1.3$ (i.e. weak negative spurious correlation). As shown in Figure 2, the training data is the union of two sub-spaces. In testing, we vary $r \in \{-1.5, -1.7, -1.9, -2.3, -2.7, -3.0\}$ to simulate stronger negative spurious correlations between $V$ and $Y$. Notably, the testing data also lie on the same manifold as the training. We use the *linear model* and calculate the root-mean-square errors (RMSE) and the parameter estimation errors Est Error $= \|\hat{\theta} - \theta^*\|_2$ of different methods ($\theta^* = [\theta_S, 0, \ldots, 0]^T$). The results are shown in the *Simulation 1* in Table 1.

**Simulation Setting 2**   Then to test whether GDRO could resist label noises, we randomly sample 20 points and add label noises to them via $\tilde{Y} = Y + \text{Std}(Y)$ where std$(Y)$ denotes the standard derivation of the marginal distribution of $Y$. The results are shown in the *Simulation 2* in Table 1. And we visualize the learned worst-case distribution of three methods in Figure 3(a) and 3(b).

**Analysis**   **(1)** From the results of *Simulation 1* and *Simulation 2* in Table 1, GDRO outperforms all the baselines in terms of low prediction error on the minor group under different strengths of spurious correlations. **(2)** From *Simulation 2* in Table 1, compared with KL-DRO and $\chi^2$-DRO, GDRO is only slightly affected by the label noises. Also, from Figure 3(a), compared with GDRO, KL-DRO puts much heavier weights on the noisy points (red points of $f$-DRO are much larger). And GDRO focuses more on the minor group (blue points), which results in their different performances under *Simulation 2*. Further, to investigate this phenomenon, we quantify the smoothness via Dirichlet Energy. In Figure 3(b), we plot the Dirichlet Energy w.r.t the relative entropy $KL(\hat{P}\|\hat{P}_{tr})$ between the learned distribution $\hat{P}$ and training distribution $\hat{P}_{tr}$, which proves that the learned weights of GDRO are much smoother w.r.t. the data manifold. And this property helps GDRO to resist the label noises, since GDRO does not allow extremely high weights on the isolated points. **(3)** The third sub-figure in Figure 3(a) verifies our analysis on WDRO that it introduces much more label noises (red points).

**Discussion on kNN**   To test whether GDRO is sensitive to the parameter $k$ of the kNN graph $G_0$, we vary $k \in \{5, 20, 100\}$ and test the performances of our GDRO under *simulation setting 1*. We also visualize the kNN graphs in Figure 2, which show that kNN consistently manages to fit the data manifold well until $k = 100$. And empirical results of *Simulation 3* in Table 1 prove that with $k < 100$, GDRO performs stably better than the baselines with small and moderate $k$, except that smaller $k$ leads to slower convergence since sparse graphs restrain the flow of probability weights. Still, we present an extreme *failure case* where KNN achieves a poor approximation of the data manifold. When $k$ increases to an extremely large number as $k = 100$, the neighborhood of kNN diffuses and two manifolds start to merge on the graph, in which case GDRO could not distinguish between two sub-populations and its performance degrades as shown in the Table 1. Actually, in Theorem 3.4 of this paper, we have proved that with an infinitely large $k$, GDRO could be reduced to KL-DRO, which completely ignores data geometry. Still, we have to clarify that kNN and GDRO perform stably well for a large range of $k$.

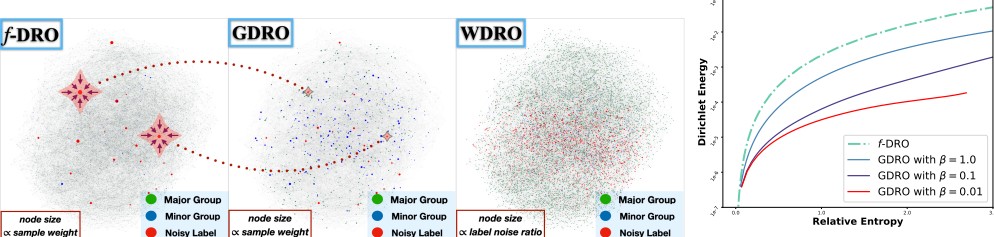

(a) Visualization of learned weights of $f$-DRO and GDRO.          (b) Weight smoothness.

Figure 3: Explanatory studies of *Simulation 2* for the regression data. **Figure (a)** visualizes the learned worst-case distribution of $f$-DRO, GDRO, and WDRO on kNN, and the size of each node is proportional to its sample weight or its label noise ratio. **Figure (b)** plots the Dirichlet Energy w.r.t the relative entropy, which measures the smoothness of learned weights given the same $D_{KL}(p\|\hat{P}_{tr})$.

## 2. Classification: Sub-population Shifts with High-dimensional Manifold Data

**Data Generation**   In this setting, data are high-dimensional but with a low-dimensional structure. The data generation is similar to [25] and is a typical classification setting in OOD generalization.

We introduce the spurious correlation between the label $Y = \{+1, -1\}$ and the spurious attribute $A = \{+1, -1\}$. We firstly generate low-dimensional data $X_{low} = [S, V]^T \in \mathbb{R}^{10}$ as:

$$S \sim \mathcal{N}(Y\mathbf{1}, \sigma_s^2 \mathbb{I}_5), \quad V \sim \mathcal{N}(A\mathbf{1}, \sigma_v^2 \mathbb{I}_5), \quad \text{where } A = \begin{cases} Y, & \text{with probability } r, \\ -Y, & \text{with probability } 1 - r. \end{cases} \quad (15)$$

Intuitively, $r \in [0, 1]$ tunes the proportions of sub-populations and controls the spurious correlation between $A$ and $Y$. When $r > 0.5$, the spurious attribute $A$ is positively correlated with $Y$; and when $r < 0.5$, the spurious correlation becomes negative. And larger $|r - 0.5|$ results in stronger spurious correlation between $A$ and $Y$. Then to convert the low-dimensional data to high-dimensional space, $X_{low}$ is multiplied by a column full rank matrix $H$ as:

$$X_{high} = (HX_{low}) \in \mathbb{R}^{300}, \quad (16)$$

where $H \in \mathbb{R}^{300 \times 10}$ is full column rank, and we randomly choose $H$ in each run.

**Simulation Setting** For both the training and testing data, we set $\sigma_s^2 = 1.0$ and $\sigma_v^2 = 0.3$. We use *linear models* with cross-entropy loss for all methods. In training, we set $r = 0.85$ ($A$ is positively correlated with $Y$). In testing, we design two environments with $r_1 = 0.5$ ($A \perp Y$) and $r_2 = 0.0$ ($A$ is negatively correlated with $Y$) to introduce distributional shifts. Apart from the natural setting without label noises, we also test the performances under label noises. Specifically, we add 4% label noises in the training data by flipping the label $Y$. We run the experiments 10 times, each time with one random matrix $H$. We report the mean accuracy in Table 2.

**Analysis** From the results in Table 2, our GDRO outperforms all baselines under the sub-population shifts, and it is not affected much by the label noises, which validates the effectiveness of our GDRO.

## 4.2 Real-World Data

We evaluate our method on four real-world datasets. Due to space limits, we place two of them here, with various kinds of distributions, including sub-populations shifts and class difficulty shifts, and the others can be found in Appendix. We use MLPs with cross-entropy loss in these experiments.

**Colored MNIST: Sub-population Shifts & Label Noises** Following Arjovsky *et al.* [1], Colored MNIST is a binary classification task constructed on the MNIST dataset. Firstly, a binary label $Y$ is assigned to each image according to its digit: $Y = 0$ for digit $0 \sim 4$ and $Y = 1$ for digit $5 \sim 9$. Secondly, we induce noisy labels $\tilde{Y}$ by randomly flipping the label $Y$ with a probability of 0.2. Then we sample the color id $C$ spuriously correlated with $\tilde{Y}$ as $C = \begin{cases} +\tilde{Y}, & \text{with probability } 1 - r, \\ -\tilde{Y}, & \text{with probability } r. \end{cases}$

Intuitively, $r$ controls the spurious correlation between $Y$ and $C$. When $r < 0.5$, $C$ is positively correlated with $Y$; and when $r > 0.5$, the spurious correlation becomes negative. And $|r - 0.5|$ controls the strength of the spurious correlation. In *training*, we randomly sample 5000 data points and set $r = 0.85$ (*strong negative* spurious correlation between $C$ and $Y$) and in *testing*, we set $r = 0$ (*strong positive* spurious correlation), inducing strong shifts between training and testing. Results are shown in Table 3.

**Ionosphere Radar Classification: Class Difficulty Shifts** Ionosphere Radar Dataset [8] consists of return signals from the ionosphere of a phased array radar system in Google Bay, Labrador. The electromagnetic signals were processed by an auto-correlation function to produce 34 continuous attributes. The task is to predict whether the return signal indicates specific physical structures in the ionosphere (good return) or not (bad return). However, the prediction difficulty of two classes is quite different, and ERM was found to achieve a much lower accuracy on bad returns than good ones [28]. In this experiment, both the *training and testing* sets consist of samples with balanced label distribution. But due to the disparity of class difficulty, the prediction accuracy of two classes is quite different, while DRO methods are expected to achieve similar prediction accuracy for both classes. Therefore, in *testing*, we report the testing accuracy for the easy class and the hard class respectively, as well as the AUC score of the testing set. Results are shown in Table 3.

**Analysis** From the results on real-world data, we find that all DRO methods (WDRO and $f$-DROs) show significant promotions to ERM, reflecting the reasonability of our experimental settings. And our proposed GDRO outperforms all baselines significantly when dealing with sub-population shifts and class difficulty shifts, which validates the effectiveness of our GDRO.

Table 2: Results of the classification simulated experiment.

|  | No Label Noises | | Add 4% Label Noises | |
|---|---|---|---|---|
|  | $r_1 = 0.5$ | $r_2 = 0.0$ | $r_1 = 0.5$ | $r_2 = 0.0$ |
| ERM | 0.573 | 0.153 | 0.573 | 0.152 |
| WDRO | 0.576 | 0.159 | 0.576 | 0.157 |
| KL-DRO | 0.654 | 0.340 | 0.625 | 0.269 |
| $\chi^2$-DRO | 0.734 | 0.644 | 0.666 | 0.554 |
| GDRO | **0.768** | **0.767** | **0.760** | **0.703** |

Table 3: Results of Colored MNIST data and Ionosphere data.

| Method | Colored MNIST | | Ionosphere | | |
|---|---|---|---|---|---|
|  | Train Acc | Test Acc | Easy Class Acc | Hard Class Acc | AUC Score |
| ERM | 0.867 | 0.116 | 0.952 | 0.481 | 0.683 |
| WDRO | **1.000** | 0.335 | 0.944 | 0.630 | 0.774 |
| $\chi^2$-DRO | 0.839 | 0.420 | 0.976 | 0.519 | 0.756 |
| KLDRO | **1.000** | 0.287 | **0.984** | 0.630 | 0.826 |
| GDRO | 0.717 | **0.696** | 0.962 | **0.741** | **0.883** |

## 5   Related Work

Distributionally robust optimization (DRO) directly solves the OOD generalization problem by optimizing the worst-case error in a pre-defined uncertainty set, which is often constrained by moment or support conditions [9, 3], shape constraints [22, 17, 16, 5], $f$-divergence [23, 11] and Wasserstein distance [26, 29, 4, 12]. [9, 3] set moment or support conditions for the distributions in the uncertainty set. As for shape constraints, one commonly used is unimodality, and [16] uses the orthounimodality to constitute the uncertainty set for DRO for multivariate extreme event analysis. As for $f$-divergence, [23] theoretically demonstrates that it is equivalent to the variance penalty, and [11] derives the optimization algorithm from its dual reformulation. Compared with $f$-divergences which require the support of distributions in the uncertainty set is fixed, the uncertainty set built with Wasserstein distance contains distributions with different support and could provide robustness to unseen data. Despite the capacity of a Wasserstein uncertainty set, the optimization of Wasserstein DRO is quite hard. [26, 12, 4] convert the original DRO problem into a regularized ERM problem, but it is suitable only for a limited class of loss functions and transportation cost functions. [29] proposes an approximate optimization method for Wasserstein DRO and could be applied to deep neural networks, which protects the models from adversarial attacks. Besides, DRO methods have also been used for structured data. [18] studies the DRO problem for data generated by a time-homogeneous, ergodic finite-state Markov chain.

Although DRO methods could guarantee the OOD generalization performances when the testing distribution is included in the uncertainty set, there are works [15, 13] doubting their real effects in practice. In order to guarantee the OOD generalization ability, in real scenarios, the uncertainty set has to be overwhelmingly large to contain the potential testing distributions. Such overwhelmingly large set makes the learned model make decisions with fairly low confidence, and it is also referred to as the over-pessimism problem. To mitigate such problem, [13] proposes to incorporate additional unlabeled data to further constrain the uncertainty set, and [20] learns the transportation cost function for WDRO with the help of multiple environment data.

## 6   Conclusion

Through this work, we take the first step to incorporate data geometry information to mitigate the over-flexibility problem in DRO. In this work, we use the k-nearest-neighbor graph to characterize the data manifold, while our proposed method is compatible with any manifold learning or graph learning methods. And we believe that a more accurate estimated data structure with advanced manifold learning and graph learning algorithms will further boost the performance of GDRO, which we leave for future work.

## Acknowledgements

This work was supported in part by National Key R&D Program of China (No. 2018AAA0102004, No. 2020AAA0106300), National Natural Science Foundation of China (No. U1936219, 62141607), Beijing Academy of Artificial Intelligence (BAAI). Bo Li's research was supported by the National Natural Science Foundation of China (No.72171131); the Tsinghua University Initiative Scientific Research Grant (No. 2019THZWJC11); Technology and Innovation Major Project of the Ministry of Science and Technology of China under Grants 2020AAA0108400 and 2020AAA0108403. We would like to thank Yuting Pan, Renzhe Xu, Hao Zou for helpful comments.

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
