[12, 4] set moment or support conditions for the distributions in the uncertainty set. As for shape constraints, one commonly used is unimodality, and [20] uses the orthounimodality to constitute the uncertainty set for DRO for multivariate extreme event analysis. Compared with the shape constraints, our proposed GDRO intrinsically considers the data geometric properties via Geometric Wasserstein distance, and the geometric property is learned in a data-driven way which is compatible with manifold learning and graph learning methods. As for $f$-DRO, [27] theoretically demonstrates that it is equivalent to the variance penalty, and [15] derives the optimization algorithm from its dual reformulation. Compared with $f$-divergences which require the support of distributions in the uncertainty set is fixed, the uncertainty set built with Wasserstein distance contains distributions with different support and could provide robustness to unseen data. Despite the capacity of a Wasserstein uncertainty set, the optimization of Wasserstein DRO is quite hard. [31, 16, 5] convert the original DRO problem into a regularized ERM problem, but it is suitable only for a limited class of loss functions and transportation cost functions. [34] proposes an approximate optimization method for Wasserstein DRO and could be applied to deep neural networks, which protects the models from adversarial attacks. Besides, DRO methods have also been used for structured data. For example, [22] studies the DRO problem for data generated by a time-homogeneous, ergodic finite-state Markov chain.

Although DRO methods could guarantee the OOD generalization performances when the testing distribution is included in the uncertainty set, there are works [19, 17] doubting their real effects in practice. In order to guarantee the OOD generalization ability, the uncertainty set should be large enough to capture the potential testing distribution, while in real scenarios, the uncertainty set has to be overwhelmingly large to achieve this. Such overwhelmingly large set makes the learned model make decisions with fairly low confidence, and it is also referred to as the over-pessimism problem. To mitigate such problem, [17] proposes to incorporate additional unlabeled data to further constrain the uncertainty set, and [24] learns the transportation cost function for WDRO with the help of multiple environment data. Different from these works that utilize additional information to constrain the uncertainty set, our proposed GDRO naturally incorporates the data geometric properties into the design of the uncertainty set by firstly using the new distance metric, Geometric Wasserstein distance.

Apart from DRO methods, there are also multiple branches of methods addressing the problem of OOD generalization. Domain generalization methods utilize training data from multiple domains to learn models that generalizes well to unseen domains, and for details of this branch, one can refer to [37]. Invariant learning methods [1], from the causal inference literature, assume the existence of invariant representation and leverage multiple environment data to learn such representations. Compared with DRO methods, they rely on strong assumptions and lack theoretical guarantees.

## A.2   Derivations of the Optimization

The whole Forward Euler Method in Section 3.2 is given as:

$$j_{ij} = 1/n(p_i(\ell_j - \ell_i + \beta(\log p_i - \log p_j))_+ - p_j(\ell_j - \ell_i + \beta(\log p_i - \log p_j))_-), \quad (17)$$

$$p_i^{(t+\alpha)} = p_i^{(t)} - \frac{\alpha}{2} \sum_{j:(i,j)\in E} (j_{ij} - j_{ji})w_{ij}, \quad (18)$$

where $(a)_+ = \max\{a, 0\}$ and $(a)_- = (-a)_+$.

## A.3   Proof of Theorem 3.1

**Theorem A.1** (Local Robustness Guarantees of Lagrangian Penalty Problem). *For any $t > 0$, $\tau > 0$ and given $\theta$, denote the solution of Equation 5 as $p^*(\theta) = \arg\sup_{p\in\mathscr{P}(G_0)} \mathcal{R}_n(\theta, p) -$*

$\frac{1}{2\tau}\mathcal{GW}^2_{G_0}(\hat{p}^t_\tau(\theta), p)$. Let $\epsilon_\tau(\theta) = \mathcal{GW}^2_{G_0}(\hat{p}^t_\tau(\theta), p^*(\theta))$, we have

$$\min_{\theta \in \Theta} \sup_{p \in \mathscr{P}_o(G_0)} \mathcal{R}_n(\theta, p) - \frac{1}{2\tau}\mathcal{GW}^2_{G_0}(\hat{p}^t_\tau(\theta), p) = \min_{\theta \in \Theta} \sup_{p:\mathcal{GW}^2_{G_0}(\hat{p}^t_\tau(\theta), p) \leq \epsilon_\tau(\theta)} \mathcal{R}_n(\theta, p). \tag{19}$$

*Proof.* Denote $p^* = \arg\sup_{p \in \mathscr{P}_o(G_0)} \mathcal{R}_n(\theta, p) - \frac{1}{2\tau}\mathcal{GW}^2_{G_0}(\hat{p}^t_\tau, p)$, since $\epsilon_\tau(\theta) = \mathcal{GW}^2_{G_0}(\hat{p}^t_\tau, p^*)$, here we proof by contradiction. Assume $p' = \arg\sup_{p:\mathcal{GW}^2_{G_0}(\hat{p}^t_\tau(\theta), p) \leq \epsilon_\tau(\theta)} \mathcal{R}_n(\theta, p)$, then we have $\mathcal{R}(\theta, p') \geq \mathcal{R}(\theta, p^*)$ and $\mathcal{GW}^2_{G_0}(\hat{p}^t_\tau, p') \leq \epsilon_\tau(\theta)$, and therefore $\mathcal{GW}^2_{G_0}(\hat{p}^t_\tau, p') \leq \mathcal{GW}^2_{G_0}(\hat{p}^t_\tau, p^*)$. Denote $\mathcal{L}(\theta, p) = \mathcal{R}_n(\theta, p) - \frac{1}{2\tau}\mathcal{GW}^2_{G_0}(\hat{p}^t_\tau(\theta), p)$, then we have $\mathcal{L}(\theta, p^*) \leq \mathcal{L}(\theta, p')$. Since $p^*$ is the supremum point of $\mathcal{L}(\theta, \cdot)$, it must be $\mathcal{L}(\theta, p^*) = \mathcal{L}(\theta, p')$, which gives that $\mathcal{R}(\theta, p') = \mathcal{R}(\theta, p^*)$. $\square$

## A.4 Proof of Theorem 3.2

The proof is based on the Theorem 5 in [7]. From [7], we have

$$\mathcal{R}(p^\infty) - \mathcal{R}(p(t)) \leq e^{-Ct}(\mathcal{R}(p^\infty) - \mathcal{R}(p^0)). \tag{20}$$

Then denote the real worst-case distribution within the $\epsilon(\theta)$-radius discrete Geometric Wasserstein-ball as $p^*$, that is,

$$p^* = \arg \sup_{p:\mathcal{GW}^2_{G_0}(\hat{P}_{tr}, p) \leq \epsilon(\theta)} \sum_{i=1}^n p_i \ell_i - \beta \sum_{i=1}^n p_i \log p_i, \tag{21}$$

and we have

$$\mathcal{R}(p^\infty) - \mathcal{R}(p^*) + \mathcal{R}(p^*) - \mathcal{R}(p(t)) \leq e^{-Ct}(\mathcal{R}(p^\infty) - \mathcal{R}(p^*) + \mathcal{R}(p^*) - \mathcal{R}(p^0)). \tag{22}$$

Therefore, we have

$$\mathcal{R}(p^*) - \mathcal{R}(p(t)) \leq e^{-Ct}(\mathcal{R}(p^*) - \mathcal{R}(p^0)) - (1 - e^{-Ct})(\mathcal{R}(p^\infty) - \mathcal{R}(p^*)), \tag{23}$$

and

$$\frac{\mathcal{R}(p^*) - \mathcal{R}(p(t))}{\mathcal{R}(p^*) - \mathcal{R}(p^0)} \leq e^{-Ct} - (1 - e^{-Ct})\frac{\mathcal{R}(p^\infty) - \mathcal{R}(p^*)}{\mathcal{R}(p^*) - \mathcal{R}(p^0)} < e^{-Ct}. \tag{24}$$

(choose $\hat{P}_{tr}$ as $p^0$).

## A.5 Proof of Theorem 3.3

We follow Sinha *et al.*[34] for proof of the convergence properties of our Algorithm. Denote the worst-case distribution as:

$$p^*(\theta) = \arg \max_{p:\mathcal{GW}^2_{G_0}(p, \hat{P}_{tr}) \leq \hat{\epsilon}(\theta)} \sum_{i=1}^n p_i \ell(f_\theta(x_i), y_i), \tag{25}$$

and our objective function as:

$$F(\theta) = \sum_{i=1}^n p^*_i(\theta)\ell(f_\theta(x_i), y_i), \tag{26}$$

and our learned distribution after $k$ times gradient flow is denoted as $p^k$. The gradient descent of $\theta$ is:

$$\theta^{(t+1)} = \theta^{(t)} - \alpha_t \cdot g^{(t)}, \tag{27}$$

where $g^{(t)} = \nabla_\theta(\sum_{i=1}^n \hat{p}^*_i \ell(f_\theta(x_i), y_i))$ is the gradient approximately calculated under our learned distribution $\hat{p}^*$.

By a Taylor expansion using the $L$-smoothness of the objective $F$, we have:

$$F(\theta^{(t+1)}) \leq F(\theta^{(t)}) + \langle \nabla_\theta F(\theta^{(t)}), \theta^{(t+1)} - \theta^{(t)} \rangle + \frac{L}{2}\|\theta^{(t+1)} - \theta^{(t)}\|^2_2 \tag{28}$$

$$= F(\theta^{(t)}) - \alpha\langle \nabla_\theta F(\theta^{(t)}), g^{(t)} \rangle + \frac{L\alpha^2}{2}\|g^{(t)}\|^2_2. \tag{29}$$

Then denote the gradient error as:

$$\delta^{(t)} = \nabla_\theta F(\theta^{(t)}) - g^{(t)}, \tag{30}$$

thus, $g^{(t)} = \nabla_\theta F(\theta^{(t)}) - \delta^{(t)}$, and we have:

$$F(\theta^{(t+1)}) \leq F(\theta^{(t)}) - \alpha \langle \nabla_\theta F(\theta^{(t)}), g^{(t)} \rangle + \frac{L\alpha^2}{2} \|g^{(t)}\|_2^2 \tag{31}$$

$$\leq F(\theta^{(t)}) - \alpha \langle \nabla_\theta F(\theta^{(t)}), \nabla_\theta F(\theta^{(t)}) - \delta^{(t)} \rangle + \frac{L\alpha^2}{2} \|\nabla_\theta F(\theta^{(t)}) - \delta^{(t)}\|_2^2 \tag{32}$$

$$= F(\theta^{(t)}) - \alpha \|\nabla_\theta F(\theta^{(t)})\|_2^2 + \alpha \langle \nabla_\theta F(\theta^{(t)}), \delta^{(t)} \rangle + \frac{L\alpha^2}{2} \|\nabla_\theta F(\theta^{(t)}) - \delta^{(t)}\|_2^2 \tag{33}$$

$$\leq F(\theta^{(t)}) - \frac{\alpha}{2} \|\nabla_\theta F(\theta^{(t)})\|_2^2 + \frac{\alpha}{2} \|\delta^{(t)}\|_2^2 + L\alpha^2 \left( \|\nabla_\theta F(\theta^{(t)})\|_2^2 + \|\delta^{(t)}\|_2^2 \right) \tag{34}$$

$$= F(\theta^{(t)}) - \frac{\alpha}{2}(1 - 2L\alpha)\|\nabla_\theta F(\theta^{(t)})\|_2^2 + \frac{\alpha}{2}(1 + 2L\alpha)\|\delta^{(t)}\|_2^2. \tag{35}$$

Therefore, we have:

$$(1 - 2L\alpha)\|\nabla_\theta F(\theta^{(t)})\|_2^2 - (1 + 2L\alpha)\|\delta^{(t)}\|_2^2 \leq \frac{2}{\alpha} \left( F(\theta^{(t)}) - F(\theta^{(t+1)}) \right), \tag{36}$$

and average from $t = 0$ to $K$ we have:

$$(1 - 2L\alpha)\frac{1}{K}\sum_{t=1}^{K} \|\nabla_\theta F(\theta^{(t)})\|_2^2 - (1 + 2L\alpha)\frac{1}{K}\sum_{t=1}^{K} \|\delta^{(t)}\|_2^2 \leq \frac{2}{\alpha K} \left( F(\theta^{(0)}) - F(\theta^{(K+1)}) \right), \tag{37}$$

and

$$\frac{1}{K}\sum_{t=1}^{K} \|\nabla_\theta F(\theta^{(t)})\|_2^2 - \frac{(1 + 2L\alpha)}{1 - 2L\alpha}\frac{1}{K}\sum_{t=1}^{K} \|\delta^{(t)}\|_2^2 \leq \frac{2}{\alpha K(1 - 2L\alpha)} \left( F(\theta^{(0)}) - F(\theta^{(K+1)}) \right), \tag{38}$$

Take expectations on the both sides like:

$$\frac{1}{K}\mathbb{E}\left[\sum_{t=1}^{K} \|\nabla_\theta F(\theta^{(t)})\|_2^2\right] - \frac{(1 + 2L\alpha)}{1 - 2L\alpha}\frac{1}{K}\sum_{t=1}^{K} \|\delta^{(t)}\|_2^2 \leq \frac{2}{\alpha K(1 - 2L\alpha)} \left( F(\theta^{(0)}) - \mathbb{E}\left[F(\theta^{(K+1)})\right] \right). \tag{39}$$

Then we only have to bound the $\delta^{(t)}$. Following Sinha *et al.*[34], we deal with $\|\delta^{(t)}\|_2^2$. According to the assumption on $R(\theta, p)$, we have

$$\|\delta^{(t)}\|_2^2 = \|\nabla_\theta F(\theta^{(t)}) - g^{(t)}\|_2^2 = \|\nabla_\theta R(\theta^{(t)}, p^*) - \nabla_\theta R(\theta^{(t)}, \hat{p}^*)\|_2^2 \tag{40}$$

$$\leq L_p^2 \|p^* - \hat{p}^*\|_2^2 \tag{41}$$

$$\leq L_p^2 \gamma. \tag{42}$$

### A.6 Proofs of Theorem 3.4

When relaxing the constrains of $\epsilon$-radius Geometric Wasserstein ball, the objective function becomes:

$$\min_{\theta \in \Theta} \sup_{p \in \mathscr{P}_o(G_0)} \left\{ \mathcal{R}_n(\theta, p) = \sum_{i=1}^{n} p_i \ell_i - \beta \sum_{i=1}^{n} p_i \log p_i \right\}, \tag{43}$$

which is equivalent to

$$\min_{\theta \in \Theta} \sup_{p \in \mathscr{P}_o(G_0)} \sum_{i=1}^{n} p_i \ell_i - \beta \cdot \mathrm{D}_{KL}(p \| \hat{P}_{tr}). \tag{44}$$

Then it naturally gives the results in Theorem 3.4 (the proof is similar to Theorem 3.1 and we omit here).

## A.7 Relations between GDRO and KL-DRO

Apart from Theorem 3.4, we introduce a more straightforward proposition showing the equivalence of KL-DRO and GDRO as the entropy regularization $\beta \to \infty$.

**Proposition A.1** (Reduction of GDRO to KL-DRO). *The objective function of GDRO in Equation (4) is equivalent to the following objective of KL-DRO as $\beta \to \infty$:*

$$\min_{\theta \in \Theta} \left\{ G_n^{KL}(\theta) = \sup_{P: D_{KL}(P \| \hat{P}_{tr}) \leq \hat{\epsilon}(\theta)} \sum_{i=1}^n p_i \ell(f_\theta(x_i), y_i) \right\}, \quad \text{with } \hat{\epsilon}(\theta) = D_{KL}(p^*(\theta) \| \hat{P}_{tr}), \tag{45}$$

*where $p^*(\theta) = \arg\max_p \sum_{i=1}^n p_i \ell(f_\theta(x_i), y_i) - \beta \sum_{i=1}^n p_i \log p_i$.*

*Proof.* We prove the proposition by showing that both the objectives of GDRO and KL-DRO are reduced to an ERM objective for infinitely large $\beta$:

$$\min_{\theta \in \Theta} \left\{ G_n^{ERM}(\theta) = \frac{1}{n} \sum_{i=1}^n \ell(f_\theta(x_i), y_i) \right\}. \tag{46}$$

Rewrite the objective of GDRO in Equation (4) as:

$$\min_{\theta \in \Theta} \left\{ G_n^{GDRO}(\theta) = \sup_{P: \mathcal{GW}_{G_0}^2(\hat{P}_{tr}, P) \leq \epsilon} \sum_{i=1}^n p_i \ell(f_\theta(x_i), y_i) - \beta \sum_{i=1}^n p_i \log p_i \right\}. \tag{47}$$

For any $P \in \mathscr{P}_0(G_0)$ and $P \neq P^U$ where $P^U = (\frac{1}{n}, ..., \frac{1}{n})$ is a uniform distribution, since $\sum_{i=1}^n p_i \log p_i > \sum_{i=1}^n p_i^U \log p_i^U = -\log n$ and $\sum_{i=1}^n p_i \ell(f_\theta(x_i), y_i)$ is bounded w.r.t. $P$, there exists $\beta_0$ such that for any $\beta > \beta_0$:

$$\sum_{i=1}^n p_i \ell(f_\theta(x_i), y_i) - \beta \sum_{i=1}^n p_i \log p_i < \sum_{i=1}^n \frac{1}{n} \ell(f_\theta(x_i), y_i) + \beta \log n. \tag{48}$$

Therefore, as $\beta \to \infty$,

$$\sup_{P: P \in \mathscr{P}_0(G_0)} \sum_{i=1}^n p_i \ell(f_\theta(x_i), y_i) - \beta \sum_{i=1}^n p_i \log p_i = \sum_{i=1}^n \frac{1}{n} \ell(f_\theta(x_i), y_i) + \beta \log n. \tag{49}$$

The supremum is achieved at $P = P^U$. Since $P^U$ satisfies $\mathcal{GW}_{G_0}^2(\hat{P}_{tr}, P) = 0 \leq \epsilon$ for any positive $\epsilon$, the objective of GDRO is reduced to:

$$G_n^{GDRO}(\theta) = \frac{1}{n} \sum_{i=1}^n \ell(f_\theta(x_i), y_i) + \beta \log n, \quad \text{as } \beta \to \infty, \tag{50}$$

which is equivalent as $G_n^{ERM}$ except for a constant independent of $\theta$.

Next, the objective of KL-DRO $G_n^{KL}(\theta)$ could be similarly reformulated as is in the proof of Theorem 3.1:

$$G_n^{KL}(\theta) = \sup_{P: P \in \mathscr{P}_o(G_0)} \sum_{i=1}^n p_i \ell(f_\theta(x_i), y_i) - \beta \sum_{i=1}^n p_i \log p_i. \tag{51}$$

According to Equation 49, as $\beta \to \infty$,

$$G_n^{KL}(\theta) = \sum_{i=1}^n \frac{1}{n} \ell(f_\theta(x_i), y_i) + \beta \log n = G_n^{GDRO}(\theta). \tag{52}$$

$\square$

## A.8 Limitations

GDRO handles unseen inside-manifold distributions with various categories of shifts as is stated in the experiment section, including sub-population shifts and class difficulty shifts. However, generalization to target data entirely falling out of the training data's manifold, known as support shift or non-overlapping support, is intrinsically a hard problem. In standard supervised learning, covariate shift with arbitrary support is known to be intractable [3]. Some domain adaptation methods, such as invariant representation learning [39], could empirically validate its effectiveness out of support while still requiring the target distribution to be close to the source's. Not to mention that such methods have utilized unlabeled target data from the unseen support. Therefore, out-of-support generalization is not the focus of GDRO for lack of additional information and strong structural assumptions. And we leave it to future work.

## B Experiments

In this section, we introduce the details of our experiments.

**Dataset Summary**   In order to comprehensively evaluate the empirical performance of our proposed GDRO, we experiment on both simulated and real-world datasets with various distributional shift patterns studied by OOD generalization, including sub-population shifts and label shifts. The descriptions of our adopted datasets are shown in Table 4.

Table 4: Datasets descriptions.

| Dataset | Toy Example | Manifold | Selection Bias | Colored MNIST | Retiring Adults | HIV | IonoSphere |
|---|---|---|---|---|---|---|---|
| Kind | Regression | Classification | Regression | Classification | Classification | Classification | Classification |
| Data Generation | Simulation | Simulation | Simulation | Real | Real | Real | Real |
| Dimension. | 2 | 300 | 10 | 2352 | 10∼19 | 160 | 34 |
| Shift Pattern | Sub-population | Domain Shift | Domain Shift | Domain Shift | Sub-population | Label Shift | Label Shift |
| Model | Linear | Linear | Linear | MLP | Linear | MLP | MLP |

**Baselines**   We compare our GDRO with the following baselines:

- Empirical Risk Minimization (ERM):

$$\min_{\theta} \frac{1}{N} \sum_{i=1}^{N} \ell(y_i, f_\theta(x_i)). \tag{53}$$

- Wasserstein DRO (WDRO [5, 34]):

$$\min_{\theta} \sup_{Q:W_c(Q,\hat{P}_N)\leq\epsilon} \mathbb{E}_Q[\ell(Y, f_\theta(X))]. \tag{54}$$

- KL-DRO [15]:

$$\min_{\theta} \sup_{q:\mathrm{D}_{KL}(q\|\hat{P}_N)\leq\epsilon} \sum_{i=1}^{N} q_i\ell(y_i, f_\theta(x_i)). \tag{55}$$

- $\chi^2$-DRO [27, 15]:

$$\min_{\theta} \sup_{q:\mathrm{D}_{\chi^2}(q,\hat{P}_N)\leq\epsilon} \sum_{i=1}^{N} q_i\ell(y_i, f_\theta(x_i)). \tag{56}$$

- Environment Inference for Invariant Learning (EIIL [8]): this method belongs to invariant learning. As a general OOD generalization method from another branch, we temporarily add it only in the Colored MNIST experiment.

**Implementation Details**   For all experiments, $G_0$ is constructed as a k-nearest neighbor graph from the *training data only* at the initialization step. Specifically, we adopt NN-Descent to estimate the k-nearest neighbor graph for the large-scale dataset Colored MNIST while performing exact search for k-nearest neighbors in other experiments. We adopt MSE as the empirical loss function for regression tasks and cross-entropy for classification tasks. The parameterized model $f_\theta$ is implemented as

a MLP with a hidden layer of 64 neurons for the HIV and IonoSphere dataset, a MLP with two hidden layers of 128 neurons for Colored MNIST, and a linear model in the other experiments. Note that GDRO is *compatible with any parameterized models including DNN*. The training of MLP is performed with a batch size of 1024 and a learning rate at 0.001. The simulation of gradient flow in Equation 17-18 is implemented by message propagation with DGL package [38], which scales linearly with sample size and enjoys parallelization by GPU.

## B.1 Simulation Data

As for the simulation data, we simulate domain shifts between training data and testing data for both regression and classification settings. And we also investigate the influence brought by label noises to demonstrate that our GDRO could to some extent resist label noise.

### 1. Toy Example: Sub-population Shifts via Anti-Causal Effect

Firstly, inspired by [1], we induce the domain shifts via the anti-causal effect as follows:

$$S \sim \mathcal{N}(0,1), \quad Y = \alpha_S S + S^2 + \mathcal{N}(0,0.1), \quad V = \alpha_V Y + \mathcal{N}(0,1), \quad \alpha_V = \begin{cases} 1 & \textit{with probability } 1-r, \\ -0.1 & \textit{with probability } r, \end{cases}$$
(57)

where $X = [S, V]^T$, $S$ serves as a stable feature with an unchanged relationship with $Y$ (thus it should be used for prediction), but $V$ is a spurious feature with changeable relationships with $Y$ (thus one should avoid using it), and $\alpha_S$ is set to 5.0 for all data. For training data, we sample 10000 points, and design different settings with varying minor group ratios $r$. For testing data, we simulate 6 domains with strong shifts by varying $\alpha_V \in \{-3, -2, -1, 1, 2, 3\}$ ($V = \alpha_V Y + \mathcal{N}(0,1)$) and calculate the Mean_Error, Std_Error and parameter estimation error Est_Error for each method as:

- Mean Error: Mean_Error $= \frac{1}{|\mathcal{E}_{test}|} \sum_e \mathcal{L}^e$

- Standard Deviation of Error: Std_Error $= \sqrt{\frac{1}{|\mathcal{E}_{test}|-1} \sum_e (\mathcal{L}^e - \text{Mean\_Error})^2}$

- Parameter Estimation Error: Est_Error $= |\hat{\alpha}_S - \alpha_S| + |\hat{\alpha}_V|$, where $\hat{\alpha}_S$, $\hat{\alpha}_V$ denote the estimated parameters for $S$ and $V$ respectively. Note that the ground-truth $\alpha_V$ is 0.

**Analysis**    From the results in Table 5, WDRO performs similarly to ERM, and two $f$-DRO methods (KL-DRO and $\chi^2$-DRO) outperform ERM and WDRO. Such results verify our analysis that the Wasserstein uncertainty set cannot be large due to its over-flexibility, which greatly impairs its performance under strong domain shifts. Our GDRO outperforms all baselines and achieves much lower prediction errors and estimation errors in all settings, which shows our uncertainty set built on Geometric Wasserstein distance is much more practical and reasonable. Further, we plot the training data in Figure 8. From Figure 8, we can see that although the training data is low-dimensional, they have geometric structures, which are utilized by our GDRO to achieve good OOD generalization performances.

### 2. High-dimensional Data with Low-dimensional Structure: Sub-population Shifts

In this setting, data are high-dimensional but with low-dimensional structure. The data generation is similar to [30] and is a typical classification setting in OOD generalization. We introduce the spurious correlation between the label $Y = \{+1, -1\}$ and the spurious attribute $A = \{+1, -1\}$. We firstly generate low-dimensional data $X_{low} = [S, V]^T \in \mathbb{R}^{10}$ as:

$$S \sim \mathcal{N}(Y\mathbf{1}, \sigma_s^2 \mathbb{I}_5), V \sim \mathcal{N}(A\mathbf{1}, \sigma_v^2 \mathbb{I}_5),$$
(58)

Table 5: Results of the toy example with varying minor probability $r$. Each result is averaged over 10 runs, and the standard deviation is omitted since it is small for all methods.

| | **Simulation 1: Toy Example** (Domain Shift via Anti-Causal Effect) | | | | | | | | |
|---|---|---|---|---|---|---|---|---|---|
| Minor Probability | $r = 0.01$ | | | $r = 0.05$ | | | $r = 0.1$ | | |
| | Mean_Error | Std_Error | Est_Error | Mean_Error | Std_Error | Est_Error | Mean_Error | Std_Error | Est_Error |
| ERM | 7.144 | 4.260 | 3.999 | 5.594 | 3.210 | 2.970 | 4.521 | 2.444 | 2.230 |
| WDRO | 5.396 | 3.480 | 3.024 | 4.303 | 2.656 | 2.297 | 3.451 | 1.974 | 1.700 |
| KL-DRO | 2.672 | 1.145 | 1.134 | 2.678 | 1.148 | 1.194 | 2.531 | 1.086 | 1.157 |
| $\chi^2$-DRO | 3.027 | 1.323 | 1.185 | 2.954 | 1.262 | 1.118 | 2.738 | 1.079 | 0.953 |
| GDRO | **1.759** | **0.047** | **0.287** | **1.718** | **0.066** | **0.043** | **1.769** | **0.153** | **0.072** |

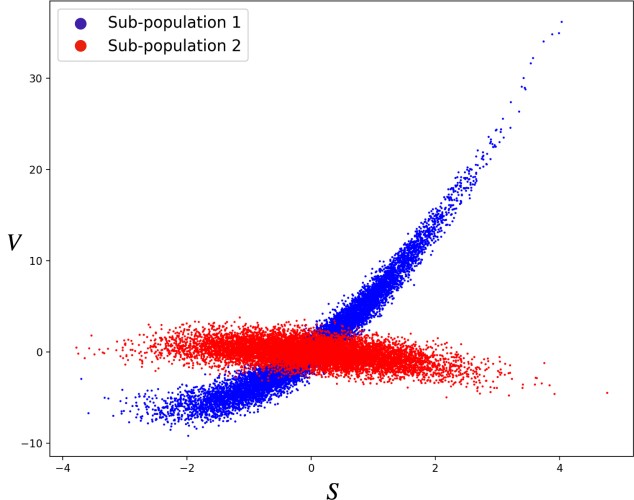

Figure 4: Data distribution of the toy example with $r = 0.5$ (the two sub-populations are balanced).

Table 6: The testing accuracy over 10 runs for the simulated experiments on high-dimensional data.

|  | No Label Noises | | Add 4% Label Noises | |
| --- | --- | --- | --- | --- |
|  | $r_1 = 0.5$ | $r_2 = 0.0$ | $r_1 = 0.5$ | $r_2 = 0.0$ |
| ERM | 0.573 | 0.153 | 0.573 | 0.152 |
| WDRO | 0.576 | 0.159 | 0.576 | 0.157 |
| KL-DRO | 0.654 | 0.340 | 0.625 | 0.269 |
| $\chi^2$-DRO | 0.734 | 0.644 | 0.666 | 0.554 |
| GDRO | **0.768** | **0.767** | **0.760** | **0.703** |

and

$$A = \begin{cases} Y, \text{with probability } r, \\ -Y, \text{with probability } 1-r. \end{cases} \tag{59}$$

Intuitively, $r \in [0, 1]$ controls the spurious correlation between $A$ and $Y$. When $r > 0.5$, the spurious attribute $A$ is positively correlated with $Y$, and when $r < 0.5$, the spurious correlation becomes negative. And larger $|r - 0.5|$ results in stronger spurious correlation between $A$ and $Y$.

Then to convert the low-dimensional data to high-dimensional space, $X_{low}$ is multiplied by a column full rank matrix $H$ as:

$$X_{high} = (HX_{low}) \in \mathbb{R}^{300}, \tag{60}$$

where $H \in \mathbb{R}^{300 \times 10}$ and each column of $H$ is linearly independent from each other ($H$ is full column rank). We randomly choose such $H$ in each run to introduce some randomness.

For the both training and testing data, we set $\sigma_s^2 = 1.0$ and $\sigma_v^2 = 0.3$. In training, we set $r = 0.85$ ($A$ is positively correlated with $Y$). In testing, we design two environments with $r_1 = 0.5$ ($A \perp Y$) and $r_2 = 0.0$ ($A$ is negatively correlated with $Y$) to introduce distributional shifts.

Apart from the natural setting without label noises, we also test the performances under label noises. Specifically, we add 4% label noises in the training data by flipping the label $Y$. We run the experiments for 10 times, and each time with one random matrix $H$. The results over 10 runs are shown in Table 6.

### 3. Selection Bias: Domain Shift via Selection Bias Mechanism

Secondly, a more complicated mechanism is designed via selection bias as:

$$S \sim \mathcal{N}(0, 2\mathbb{I}_{n_s}) \in \mathbb{R}^5, \ V \sim \mathcal{N}(0, 2\mathbb{I}_{n_v}) \in \mathbb{R}^5, \ Y = \beta^T S + 0.1 \cdot S_1 S_2 S_3 + \mathcal{N}(0, 0.5). \tag{61}$$

Similar to the toy example above, $S$ are stable features while the relationships between $V$ and $Y$ are perturbed in different domains. Specifically, a data point is selected with probability $P(x_i, y_i) = |r|^{-5*|y_i - \text{sign}(r) \cdot v_i^b|}$, which induces the spurious correlation between a certain covariate $V^b \in V$ and

$Y$. In training, we generate 10000 points, where the major group contains 95% data with $r = 1.9$ and the minor group contains 5% data with $r = -1.3$. In testing, we first report the performances of the two training groups, and then we further vary $r \in \{-1.5, -1.7, -1.9, -2.3, -2.7, -3.0\}$ to simulate more challenging domain shifts that cannot be obtained by interpolation between training groups. The average results over ten runs are shown in Table 7 (simulation 2). As for sub-population shifts, GDRO achieves good tail performance at a slight sacrifice of the performance of the major group. Further, for more challenging domain shifts, GDRO significantly outperforms all baselines in all testing distributions, even though the shifts are much stronger than the training.

Table 7: Results on the Selection Bias Experiments. We report the root mean square errors.

| | Train(major) | Train(minor) | Test | | | | | | | Parameter |
|---|---|---|---|---|---|---|---|---|---|---|
| Bias Ratio $r$ | $r = 1.9$ | $r = -1.3$ | $r = -1.5$ | $r = -1.7$ | $r = -1.9$ | $r = -2.3$ | $r = -2.7$ | $r = -3.0$ | | Est_ Error |
| | | **Simulation 2: Selection Bias Experiment without Label Noises** | | | | | | | | |
| ERM | **0.339** | 0.876 | 0.892 | 0.884 | 0.864 | 0.880 | 0.843 | 0.888 | | 0.423 |
| WDRO | **0.339** | 0.877 | 0.894 | 0.885 | 0.865 | 0.882 | 0.844 | 0.890 | | 0.424 |
| $\chi^2$-DRO | 0.411 | 0.744 | 0.757 | 0.741 | 0.733 | 0.742 | 0.714 | 0.755 | | 0.367 |
| KL-DRO | 0.370 | 0.713 | 0.728 | 0.716 | 0.708 | 0.713 | 0.685 | 0.724 | | 0.319 |
| GDRO | 0.493 | **0.492** | **0.508** | **0.489** | **0.501** | **0.483** | **0.486** | **0.496** | | **0.033** |
| | | **Simulation 3: Selection Bias Experiment under Label Noises** | | | | | | | | |
| ERM | **0.335** | 0.845 | 0.885 | 0.879 | 0.874 | 0.884 | 0.882 | 0.876 | | 0.422 |
| WDRO | **0.335** | 0.896 | 0.887 | 0.880 | 0.875 | 0.886 | 0.884 | 0.877 | | 0.423 |
| $\chi^2$-DRO | 0.375 | 0.866 | 0.855 | 0.856 | 0.843 | 0.860 | 0.854 | 0.845 | | 0.408 |
| KL-DRO | 0.393 | 0.879 | 0.868 | 0.866 | 0.856 | 0.876 | 0.866 | 0.861 | | 0.391 |
| GDRO | 0.542 | **0.537** | **0.553** | **0.549** | **0.534** | **0.539** | **0.555** | **0.550** | | **0.058** |

### 3. Label Noises: Add Label Noises

Since DRO methods are risk-aware, they are prone to be affected by label noises. Though the effect of label noises cannot be eliminated due to the nature of DRO, our proposed GDRO could significantly improve the resistance of DRO methods to a minor degree of label noises. Based on the selection bias experiment, we random sample 20 points and add label noises on them via $\tilde{Y} = Y + \text{Std}(Y)$ where $\text{std}(Y)$ denotes the standard derivation of the marginal distribution of $Y$. From results shown in Table 7(simulation 3), both $f$-DRO methods are significantly affected and perform similarly to ERM under such a minor degree of label noises. And our GDRO is only slightly affected by the label noises.

Further, to demonstrate the difference between $f$-DRO, WDRO, and GDRO, for the label noise experiment, we visualize the learned worst-case distribution of three methods. **(1)** In the first two figures in Figure 5(a), we draw the learned sample weights of KL-DRO and GDRO, where red points represent the noisy samples in training data, and the size of each point is proportional to its sample weight. We can see that KL-DRO puts heavy weights on the noisy points (the red nodes are much larger), while our GDRO only slightly increases them and the weights of data in the minor group are raised, which results in the difference between their performances in the label noise experiment. The reason for such phenomenon of KL-DRO is that it ignores the data geometry, allowing for some isolated nodes with much heavier weights than surrounding nodes, which corresponds with our Theorem 3.4. Since our GDRO intrinsically naturally constrains weight learning on the data manifold, the learned weights are smooth w.r.t the data manifold. **(2)** In the third figure in Figure 5(a), the red points represent the samples, to which WDRO introduces label noises with the ratio larger than 50%, and the size of each point is proportional to its label noise ratio. We can see that although WDRO extends the support, the created samples are quite noisy and greatly harm the learning process. **(3)** To quantify this property, we measure the smoothness via Dirichlet Energy and plot the Dirichlet Energy w.r.t the relative entropy $KL(p\|\hat{P}_{tr})$ between the learned distribution and training distribution in Figure 5(b), which shows that the learned weights of GDRO are much more smooth on the data manifold than that of KL-DRO.

### B.2   Real-World Data

Finally, towards a comprehensive comparison with existing DRO methods, we evaluate our method on four real-world datasets with various kinds of distributional shifts, including sub-population shifts, domain shifts, class-wise shifts, and label noises. All experimental details can be found in Appendix.

**1. Retiring Adults: Sub-population Shifts**   The Retiring Adults dataset [13] is derived from US Census surveys, where the sub-population shifts are natural since there are geographic variations across different states. Our experiments involve three prediction tasks defined in [13], including

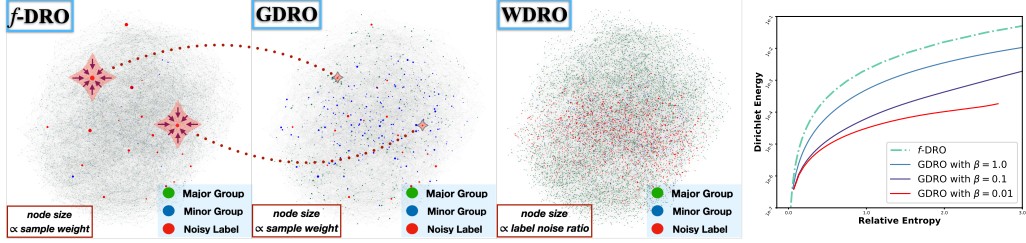

(a) Visualization of learned weights of $f$-DRO and GDRO.    (b) Weight smoothness.

Figure 5: Explanatory studies for comparison between $f$-DRO ($f(x) = x \ln x$) and GDRO. **Figure (a)** visualizes the learned worst-case distribution of $f$-DRO, GDRO, and WDRO on kNN, and the size of each node is proportional to its sample weight or its label noise ratio. **Figure (b)** plots the Dirichlet Energy w.r.t the relative entropy, which measures the smoothness of learned weights given the same $D_{KL}(p\|\hat{P}_{tr})$.

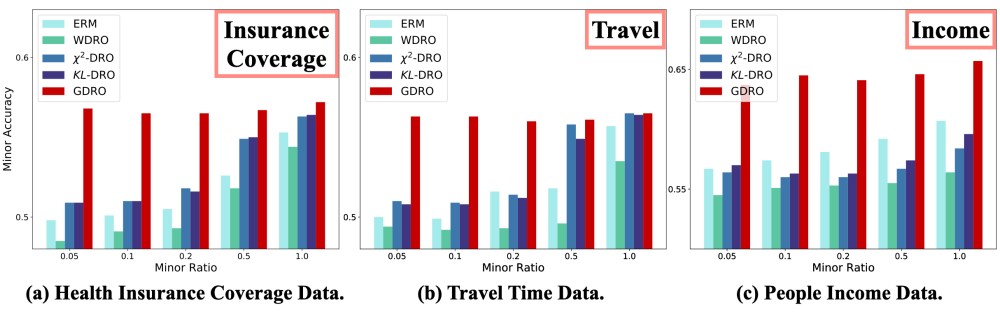

(a) Health Insurance Coverage Data.    (b) Travel Time Data.    (c) People Income Data.

Figure 6: Results of Retiring Adults Dataset.

Income Prediction, Public Health Insurance Coverage Prediction, and Commuting Time Prediction. For each task, we randomly sample 2000 points from state $A$ and $2000 \cdot r$ points from state $B$, where $r < 1$ and the second state is viewed as the minor group. In Figure 6, we report the prediction accuracy of the minor group for different minority ratios $r$ in three tasks.

**2. HIV-1: Sub-population Shifts & Label Imbalance**    HIV-1 Protease Cleavage Dataset [10] involves a task to predict whether an octamer would be cleaved by HIV-1 protease, given a 160-dimensional one-hot vector encoding the sequence of 8 amino acids composing the octamer. The dataset contains 4 splits, and following [28], we merge '746' and '1625' as subpopulations $A$ and view 'Impens' as $B$. In training, subpopulations $A$ and $B$ are mixed at a ratio of $1 : r$, where $r \leq 1$ is the minor group ratio. Coupled with the sub-population shift, class labels are imbalanced with 33% positive in sub-population $A$ and 16% positive in $B$, which is more challenging. In testing, we re-balance the two classes and plot the overall accuracy w.r.t. the minor group ratio in Figure 7.

**3. Colored MNSIT: Domain Shifts & Label Noises**    Following Arjovsky *et al.* [1], we conduct a binary classification task on the MNIST dataset. Firstly, a binary label $Y$ is assigned to each image according to its digit: $Y = 0$ for digit $0 \sim 4$ and $Y = 1$ for digit $5 \sim 9$. Secondly, we induce noisy labels $\tilde{Y}$ by randomly flipping the label $Y$ with a probability of 0.2. Then we induce domain shifts by sampling the color id $C$ spuriously correlated with $\tilde{Y}$. Specifically, we generate $C$ by flipping $\tilde{Y}$ with probability $r$, which can be viewed as the indicator of different domains. In training, we randomly sample 5000 data points and set $r = 0.85$ and in testing, we set $r = 0$, which induces strong domain shifts between training and testing. Results are shown in Table 8.

**4. Ionosphere Radar Classification: Class Difficulty Shifts**    Ionosphere Radar Dataset [11] consists of return signals from the ionosphere of a phased array radar system in Google Bay, Labrador. The electromagnetic signals were processed by an auto-correlation function to produce 34 continuous attributes. The target is to determine whether the return signal indicates specific physical structures in the ionosphere (good return) or not (bad return). Due to the disparity between classes, ERM was found to achieve a much lower accuracy on bad returns than good ones [33]. DRO methods are expected to achieve higher accuracy in the harder class. Results are shown in Table 8.

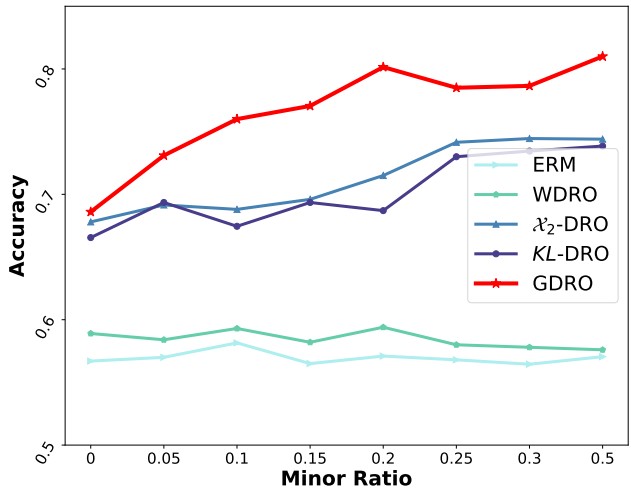

Figure 7: Results of the HIV-1 data.

Table 8: Results of Colored MNIST data and Ionosphere data.

| Method | Colored MNIST | | Ionosphere | | |
|---|---|---|---|---|---|
| | Train Acc | Test Acc | Easy Class Acc | Hard Class Acc | AUC Score |
| ERM | 0.867 | 0.116 | 0.952 | 0.481 | 0.683 |
| WDRO | **1.000** | 0.335 | 0.944 | 0.630 | 0.774 |
| $\chi^2$-DRO | 0.839 | 0.420 | 0.976 | 0.519 | 0.756 |
| KLDRO | **1.000** | 0.287 | **0.984** | 0.630 | 0.826 |
| EIIL | 0.740 | 0.596 | - | - | - |
| GDRO | 0.717 | **0.696** | 0.962 | **0.741** | **0.883** |

**Analysis** From the results on real-world data, we find that in most scenarios, WDRO only slightly outperforms ERM, and two $f$-DRO methods show significant promotions to ERM. Our proposed GDRO outperforms all baselines significantly in all scenarios. (1) The discrepancy between WDRO ($\approx$ ERM) and $f$-DRO ($>$ ERM) indicates that the extending the distribution support in practice is not as promising as might be expected, which we think is because creating new data points is nearly impossible in real scenarios. (2) The significant discrepancy between GDRO ($\gg$ ERM) and WDRO ($\approx$ ERM) shows the superiority of restricting the distribution support, which enables DRO to provide robustness in a much larger uncertainty set without worrying about generating unrealistic samples. (3) GDRO exhibits significant advantages under strong distributional shifts (small minor ratio in Figure 6 and Figure 7; strong shifts in Colored MNSIT in Table 8), which shows that by incorporating geometric properties, our uncertainty set mitigates the over-flexibility problem and our GDRO can resist stronger distributional shifts.

### B.3 Supplementary Figures

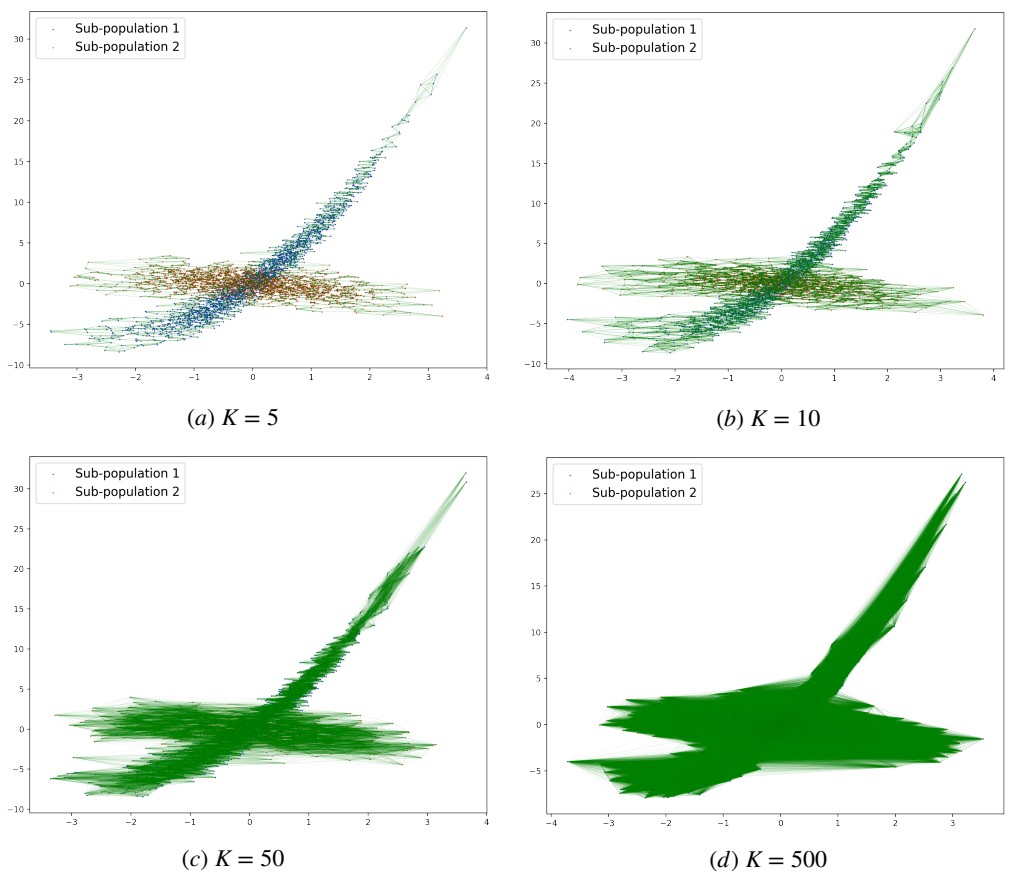

(a) $K = 5$           (b) $K = 10$

(c) $K = 50$           (d) $K = 500$

Figure 8: KNN graphs for Toy Example under various values of $K$.