# OpenReview forum: "Distributionally Robust Optimization with Data Geometry"
_NeurIPS.cc/2022/Conference — NeurIPS 2022 Accept_

### Official Review · Reviewer_aH5S · 2022-07-03

**Rating:** 7
**Confidence:** 5
**Soundness:** 3 good
**Presentation:** 3 good
**Contribution:** 3 good

**Summary:**

This work aims to solve the problem that DRO is too “pessimistic” (the uncertainty set is too large) and often leads to poor results in practice. The motivation is that “high dimensional data approximately reside on low dimensional manifolds” (lines 6-7), so this work tries to constrain the uncertainty set on this low dimensional manifold. To do this, this work (i) uses the NN-Descent method to estimate the low dimensional manifold; (ii) formulates the GDRO objective and optimizes it with alternating optimization and proves that it converges. The authors then conduct a series of experiments on synthetic and real datasets and claim that the proposed method is better than existing DRO methods.

**Questions:**

1. I suggest the authors provide some numerical results to demonstrate how well NN-Descent can estimate the low-dimensional manifold. This is very important, because if the manifold is not well estimated and the target distribution is outside the estimated manifold, then GDRO could completely fail. Moreover, since the authors are using kNN, studying the effect of k is also important. If the method works for some k but not for others, then the authors need to elaborate on how to select a proper k with the training samples alone.

2. Could the authors clarify for which of the experiments is the graph G_0 directly provided to GDRO, and for which of them is the G_0 estimated by NN-Descent?

3. In Figure 3, on the Retiring Adults dataset, GDRO seems to maintain the same high performance regardless of the minor ratio. This is not a good signal. When does the performance of GDRO start to drop? For instance, does GDRO still have a very high performance when the minor ratio is 0.01? What about when the ratio is 0? If GDRO always has such a high performance, then I believe that either G_0 provides too much information to GDRO (for instance, the target distribution could be directly obtained from G_0), or there is a bug in the code.


**Limitations:**

The limitations are not sufficiently addressed.

**Strengths And Weaknesses:**

I really like the high-level idea of this paper. “DRO is too pessimistic” is a well-known open problem in this field, and this work tries to solve this problem by constraining the uncertainty set on a low dimensional manifold, which makes lots of sense. I also think that a lot of credit should be given to the authors for the theory part: The GDRO objective is nicely formulated, and optimizing this objective with alternating optimization makes sense. The convergence result is also very nice.

The major weaknesses of this work, however, come from the experiment section. Recall that the core motivation of this work is that “high dimensional data approximately reside on low dimensional manifolds” (lines 6-7), whereas there is a huge gap between the motivation and the experiments, which makes the experiments very confusing and unconvincing.

Take the first experiment in Section 4.1 as an example. First of all, the experimental setting is very confusing. I suppose that the task is to infer Y from (S, V). I also couldn’t find the definition of alpha_V, so I suppose that it is used to define V. Thus, in this task the input data is 2-dimensional, and (S, V) seems to also reside on a 2-dimensional manifold, which is the union of a number of 1-dimensional curves (if alpha_V is in [a, b] for some a and b). I don’t think this could be called “high dimensional data approximately residing on low dimensional manifolds”.

Then the authors claim that GDRO is much better than other DRO on this task. I don’t know how the graph G_0 is estimated. I suppose that the authors just simply provide G_0 to the algorithm because NN-Descent is only used “for large-scaled datasets” (lines 109-110). So it seems to me that the reason why GDRO is so good is that G_0 leaks some additional information about the target distribution to it but not to other methods, not because it leverages the fact that the data resides on a low dimensional manifold.

Of course, it is nice that GDRO could utilize this additional leaked information from G_0. The question is: How to get this G_0 in practice? The authors propose to estimate G_0 with NN-Descent, but they don’t demonstrate how well NN-Descent can estimate G_0 on realistic tasks. If G_0 is not well estimated, and the target distribution is outside the estimated manifold, then I imagine that GDRO could completely fail.

Moreover, most of the tasks in the experiments are not really high-dimensional (<50 dimensions), and all tasks seem to follow some simple, unrealistic structures, which make it easier for GDRO to achieve high performances. It is questionable whether these good performances are transferable to real-world applications with realistic distribution shifts.

A valid experimental setting I would suggest the authors try is the following: The input data comes from a low dimensional manifold in a high dimensional space (at least 200 dimensions), but the structure of this manifold is unknown (for instance, introduce randomness into the manifold structure), so GDRO must first estimate G_0 by itself. This setting is closer to the authors’ motivation that “high dimensional data approximately reside on low dimensional manifolds”. Otherwise, it is always questionable whether the performance gain of GDRO comes from the information leakage from the provided G_0, rather than its ability to estimate and utilize the low-dimensional manifold.

In summary, I really like the high-level idea and the theory part of this paper, but the experiment section does require a lot of improvement. Currently, there is a huge gap between the authors’ motivation and the experiments, making the main conclusion of this paper highly debatable. For this reason, I recommend rejecting this paper for this time, and hope that the authors could resubmit after rewriting the experiment section.

****** Post Rebuttal ******
The authors have revised the paper as suggested, so I would like to raise my rating to accept.

---

> ### Author Response · Authors · 2022-08-02
> **Official Comment**
>
> We sincerely appreciate your approval of the idea as well as the theoretical analysis of this work and thank you for the suggestions on experiment design. Our rebuttal consists of two parts:
> * We have revised the experiment section in Appendix B based on your suggestions, and the paper will be updated accordingly if the revision turns out satisfactory;
> * We would like to address your concerns proposed in your review, which we have summarized into **6 questions**.
>
>
> ### **Q1. Source of input graph $G_0$**
> The reviewer is confused about the source of $G_0$ which is an input graph for the algorithm and raises concerns over potential information leakage of target data.
>
> We would like to **clarify** that for all experiments in this paper, $G_0$ is constructed as a k-nearest neighbor graph from the *training data only*. Thus, testing information is inaccessible to the training procedure in any form. Specifically, we adopt NN-Descent to estimate the k-nearest neighbor graph for large-scale data while performing an exact search for k-nearest neighbors in the case of small datasets. We feel sorry to have caused misunderstandings and have included detailed descriptions of implementation in the revised supplementary material in Appendix B.
>
> Therefore, the performance gain of GDRO in the shown experiments is credited to the algorithm's robustness to distributional shifts rather than its perception of target information.
>
> ### **Q2. Estimation accuracy of the data manifold**
>
> The reviewer is concerned about the estimation accuracy of the manifold with KNN and NN-Descent.
>
> We first provide numerical results of NN-Descent's accuracy for the reviewer's information: *NN-Descent typically converges to above 90% recall with each point comparing only to several percent of the whole dataset on average* [3]. As for the k-nearest neighbor graph which is either estimated by NN-Descent or exactly searched for, it is shown to have well approximated the geodesic distance within local structures on the manifold [1,2].
>
> Still, the k-nearest neighbor graph is a fundamental and basic method to represent the data structure, and manifold learning is an area with intensive research. We have to clarify that manifold learning is *not* the focus of this paper, which takes the data structure $G_0$ as input and aims at constructing a DRO objective and optimization algorithm that manages to incorporate data geometric information. Notably, since the manifold structure has long been overlooked in past DRO designs, our GDRO achieves significant performance in the experiments even with the simple KNN representation of data structure. It proves that the direction for geometric-aware DROs which we propose is valuable and promising, the uncertainty set constrained with geometric structure is reasonable, and our proposed DRO optimization method has efficiently captured the geometric information encoded in the input graph (note that no target information is leaked into $G_0$).
>
> Actually, GDRO is compatible with any manifold learning method. We do believe that a more accurate estimated data structure with advanced manifold learning algorithms will further boost the performance of GDRO, and we leave it to future work.
>
> [1] McInnes, L., Healy, J., & Melville, J. (2018). Umap: Uniform manifold approximation and projection for dimension reduction.
>
> [2] Dann, E., Henderson, N. C., Teichmann, S. A., Morgan, M. D., & Marioni, J. C. (2022). Differential abundance testing on single-cell data using k-nearest neighbor graphs.
>
> [3] Dong, W., Moses, C., & Li, K. (2011, March). Efficient k-nearest neighbor graph construction for generic similarity measures. In *Proceedings of the 20th international conference on World wide web* (pp. 577-586).

---

> > ### Author Response · Authors · 2022-08-02
> > **Q3. Demonstration of the settings of simulated experiments**
> >
> > The reviewer perceives that the settings of simulated experiments do not support the motivation of this paper which the reviewer summarizes as *high dimensional data approximately resides on low dimensional manifolds*. We would like to clarify that the manifold assumption implies that the data is situated on a hypersurface in the feature space such that it does not symmetrically stretch in each dimension, the latter of which is assumed by the uncertainty set of past DROs who ignore this geometric structure and turns out over-pessimistic. However, the manifold assumption does not enforce an absolutely high dimension of the feature space.
> >
> > In fact, we introduce the toy example in order to showcase the failure of geometric-unaware DROs in manifold data, with a low-dimensional setting for a straightforward demonstration. The description of the setting might be confusing, so we have a more detailed introduction in the revised experiment section in Appendix B. Here we would like to briefly review the setting. The feature space is spanned by $(S, V)$,  where $S$ serves as a stable(causal) feature such that the label $Y$ is generated from $S$ such that:
> > $$
> > Y =  \alpha_SS + S^2 + \mathcal{N}(0,0.1),\quad V = \alpha_V Y +\mathcal{N}(0,1)
> > $$
> > Note that $\alpha_S$ is a fixed coefficient but unknown to the algorithm, while $\alpha_V$ changes across different environments, making $V$ a spurious (anti-causal) feature because $V$ is generated by the label $Y$ in an unstable way.
> >
> > In training, $\alpha_V$ is chosen according to:
> > $$
> > \alpha_V = 1, \text{with probability }1-r;\quad  \alpha_V = -0.1, \text{with probability }r
> > $$
> >
> > Note that  $\alpha_V$ is the coefficient but unknown to the algorithm, rather than randomly distributed in an interval as the reviewer supposed (sorry for the ambiguity). Thus, the data is situated on two branches in the space spanned by $(S, V)$. The first branch corresponding to Equation 2 is centered around the parabola $V = \alpha_SS + S^2$ with Gaussian noises; the second branch corresponding to Equation 3 is centered around the parabola $V = -0.1(\alpha_SS + S^2)$. Thus, the data is approximately combined with two sub-manifolds in the feature space. The visualization of the manifold structure, which is similar to a 'cross shape', could be found in Figure 1 of Appendix B.
> >
> > In the source data where $r$ is a small positive number, such that most samples come from the first branch, a linear model trained by ERM objective tends to predict $Y$ based on $V$ to exploit the simple linear mapping in Equation 2. However, the target data is mostly sampled from the second branch where the mapping between $V$ and $Y$ significantly shifts, leading to the failure of ERM. A DRO method is supposed to capture the minority branch in the training process and to predict based on the stable feature $S$ instead of the unstable  $V$. Actually, this setting is typically adopted in the literature on OOD generalization [1,2], which contains domain shifts caused by unstable features and spurious correlation.
> >
> >
> > Existing DRO methods either create invalid samples out of the data manifold (WDRO, see Figure 2(a)) or completely ignore the geometric structure and focus on noises instead ($f$-DRO, see Figure 2(a)), both of which fail in this toy experiment (see Table 1). In contrast, our GDRO manages to capture the minority branch for being aware of the manifold structure.
> >
> > Still, it could be true that the manifold assumption is more easily satisfied in high-dimensional data. Therefore, we *follow the reviewer's suggestion to include more high-dimensional experiments in the revised version*. One of them is presented below.
> >
> > [1] Arjovsky, M., Bottou, L., Gulrajani, I., & Lopez-Paz, D. (2019). Invariant risk minimization.
> >
> > [2] Kuang, K., Xiong, R., Cui, P., Athey, S., & Li, B. (2020, April). Stable prediction with model misspecification and agnostic distribution shift.

---

> > > ### Author Response · Authors · 2022-08-02
> > > **Q4. High-dimensional simulated experiment according to the suggested valid setting**
> > >
> > > Thanks for your suggestions.
> > >
> > > #### 1. We would like to demonstrate that our Colored MNIST experiment exactly corresponds with the suggested valid setting, because
> > > * the data is high-dimensional (i.e. 2352).
> > > * the data lie on a low-dimensional manifold, which is well-accepted in computer vision.
> > >
> > > #### 2. We add one high-dimensional simulated experiment following your advice ($X_{high}\in\mathbb{R}^{300}$).
> > >
> > > The data generation process is similar to [1], which is a typical classification setting in OOD generalization. In this setting, we introduce the spurious correlation between the label $Y=\{+1,-1\}$ and $A=\{+1,-1\}$. We firstly generate low-dimensional data $X_{low}=[S,V]^T \in \mathbb{R}^{10}$ as:
> > > $$
> > > S \sim \mathcal{N}(Y{\bf 1}, \sigma_s^2\mathbb{I}_5), V \sim \mathcal{N}(A{\bf 1}, \sigma_v^2\mathbb{I}_5)
> > > $$
> > > and
> > > $$
> > > A = Y, \text{with probability }r; \quad
> > > A = -Y, \text{with probability }1-r
> > > $$
> > >
> > > Intuitively, $r$ controls the spurious correlation between $A$ and $Y$. When $r>0.5$, the spurious attribute $A$ is positively correlated with $Y$, and when $r<0.5$, the spurious correlation becomes negative.
> > > And larger $|r-0.5|$ results in stronger spurious correlation between $A$ and $Y$.
> > > Then as suggested, we convert this low-dimensional data to **high-dimensional** space via:
> > > $$
> > > X_{high} = (H X_{low}) \in \mathbb{R}^{300}
> > > $$
> > > where $H\in\mathbb{R}^{300\times 10}$ and each column of $H$ is linear independent from each other ($H$ is full column rank). We randomly choose such $H$ in each run to introduce some randomness.
> > >
> > > For the both training and testing data, we set $\sigma_s^2=1.0$ and $\sigma_v^2=0.3$.
> > > In training, we set $r=0.85$ (here $A$ is positively correlated with $Y$).
> > > In testing, we design two testing environments with $r_1=0.5$ (here $A\perp Y$) and $r_2=1.0$ (here $A$ is negatively correlated with $Y$) to introduce distributional shifts.
> > > We run the experiments 10 times, and each time with one random matrix $H$. We report the testing accuracies of the two testing environments for each method.
> > >
> > > | Without label noises                    | $r_1=0.5$            | $r_2=0.0$ |
> > > |---------------------|--------------------|---------|
> > > | ERM                 | 0.573($\pm 0.006$) |  0.153($\pm 0.011$)    |
> > > | WDRO                | 0.576($\pm 0.006$) |  0.159($\pm 0.012$)    |
> > > | KL-DRO              | 0.654($\pm 0.008$) |  0.340($\pm 0.015$)    |
> > > | $\chi^2$-DRO | 0.734($\pm 0.022$) |  0.644($\pm 0.025$)    |
> > > | GDRO                |**0.768($\pm 0.005$)** |  **0.767($\pm 0.035$)**    |
> > >
> > > Further, as done in our simulated experiments, we add label noises to the training data. Specifically, we randomly sample 200 data points (4% of the training data) and flip their labels. The results over 10 runs are as follows:
> > >
> > > | With label noises                   | $r_1=0.5$            | $r_2=0.0$ |
> > > |---------------------|--------------------|---------|
> > > | ERM                 | 0.573($\pm 0.006$) |  0.152($\pm 0.013$)    |
> > > | WDRO                | 0.575($\pm 0.008$) |  0.157($\pm 0.017$)    |
> > > | KL-DRO              | 0.625($\pm 0.007$) |  0.269($\pm 0.015$)    |
> > > | $\chi^2$-DRO | 0.666($\pm 0.037$) |  0.554($\pm 0.153$)    |
> > > | GDRO                | **0.760($\pm 0.010$)** | **0.703($\pm 0.061$)**    |
> > >
> > > From the results, our GDRO outperforms all the baselines and the performance is consistent with that under the original simulated settings.
> > > The results demonstrate that our GDRO could utilize the data geometric information to build a practical uncertainty set.
> > > Further, the second experiment with label noises also validates our analysis that GDRO could to some extent resist the influence of label noises.
> > >
> > >
> > >
> > >
> > > [1] Sagawa, S., Raghunathan, A., Koh, P. W., & Liang, P. (2020, November). An investigation of why overparameterization exacerbates spurious correlations.

---

> > > > ### Author Response · Authors · 2022-08-02
> > > > **Q5. Parameter selection for KNN**
> > > >
> > > > The reviewer suggests investigating the effect of hyperparameter $K$ in the KNN algorithm. Even though graph learning is not the focus of this paper, we show by experiments that the performances of our GDRO remain stable within a quite broad range of $K$. As is shown in the table below, the accuracy of GDRO remains strong with increasing $K$ until $K \geq 200$ when GDRO experiences performance decay but still beats the second strongest baseline $\mathcal{X}_2$-DRO. The experimental setting is the same as the manifold experiment introduced in Q4 and the reported metric is accuracy for testing data.
> > > >
> > > > | Accuracy | $K=10$           | $K=20$           | $K=50$           | $K=200$          | $K=500$          | $\mathcal{X}_2$-DRO |
> > > > | -------- | ---------------- | ---------------- | ---------------- | ---------------- | ---------------- | ------------------- |
> > > > | $r_1=0.5$  | 0.758$\pm 0.006$ | 0.768$\pm 0.005$ | 0.767$\pm 0.005$ | 0.761$\pm 0.008$ | 0.758$\pm 0.010$ | 0.734$\pm 0.022$    |
> > > > | $r_2=0.0$  | 0.753$\pm 0.036$ | 0.767$\pm 0.035$ | 0.756$\pm 0.035$ | 0.690$\pm 0.044$ | 0.669$\pm 0.047$ | 0.644$\pm 0.025$    |
> > > >
> > > > * Recall that in the training data, $r=0.85$, which means the spurious attribute $A$ is positively correlated with $Y$. And in testing, $r_1=0.5$ means $A\perp Y$, and $r_2=0.0$ means $A$ is negatively correlated with $Y$, which introduces strong distributional shifts.
> > > > * The reported testing accuracies are averaged over 10 repeated experiments.
> > > >
> > > > Thus, the hyperparameter of KNN does not require careful tuning, and any $K$ that is not excessively large suffices. Further, there exist many works analyzing how to select a proper $K$ for KNN [1,2]. In principle, KNN is designed to capture the local structure of the manifold and the neighborhood size $K$ shall not exceed the range where the local Euclidean metric holds.
> > > >
> > > > [1] Zhang, S., Li, X., Zong, M., Zhu, X., & Cheng, D. (2017). Learning k for knn classification.
> > > >
> > > > [2] Barrash, S., Shen, Y., & Giannakis, G. B. (2019, December). Scalable and adaptive KNN for regression over graphs.
> > > >
> > > >
> > > > ### **Q6 Performance on Retiring Adults dataset**
> > > > When the ratio of the minor group is too low, the performance of GDRO will drop. Here we add some results on the Health Insurance Coverage task for the Retiring Adults data.
> > > >
> > > > | Minor Group Ratio   | 0.0   | 1%    | 2.5%  |
> > > > |---------------------|-------|-------|-------|
> > > > | ERM                 | 0.485 | 0.490 | 0.490 |
> > > > | WDRO                | 0.475 | 0.477 | 0.477 |
> > > > | KL-DRO              | 0.485 | 0.496 | 0.498 |
> > > > | $\chi^2$-DRO | 0.490 | 0.497 | 0.498 |
> > > > | GDRO                | 0.476 | 0.503 | 0.527 |

---

> > > > > ### Comment · Reviewer_aH5S · 2022-08-02
> > > > > **Re: Official Comment**
> > > > >
> > > > > First of all, I want to thank the authors for this very detailed response. It is very impressive that the authors write such a detailed response in just one week. Based on this response, I would like to have further discussions with the authors:
> > > > >
> > > > > Q1-Q2. One of my main concerns is that GDRO could completely fail if the target distribution falls out of the manifold esimated by kNN, and I believe that the accuracy of kNN is crucial for GDRO. Thus, I would suggest the authors (i) use a separate paragraph to describe how kNN works, give some examples of the graph G_0, and even better provide some visualizations of the estimated manifold (e.g. for the 2-D case). (ii) Discuss possible failure cases: what could happen if the manifold is not well estimated? (iii) (Optional) Even better, talk about the sample complexity of manifold estimation by kNN. For example, how many training samples are required to estimate a 20-dimensional manifold in a 2000-dimensional space?
> > > > >
> > > > > Q3. I still have questions about the experiments. For the Simulation Data experiment, yes the training data lies on two 1-D curves (alpha_V = 1 or -0.1). However, as the authors wrote in lines 243-244, for testing data alpha_V is chosen from {-3,-2,-1,1,2,3}. This leads to two questions:
> > > > >
> > > > > (1) Could the authors clarify what is the estimated manifold in this case? If kNN can estimate the manifold very accurately as the authors claim, then the estimated manifold should be the union of the two 1-D curves, with alpha_V = 1 or -0.1. However, for the testing data, alpha_V can be 3 or -3, which falls out of the estimated manifold, so why does GDRO work? Similarly, for the Selection Bias experiment, why does GDRO work for r = -3.0 while the manifold only contains r = 1.9 and -1.3? (By the way, the setting of Selection Bias is also hard to understand)
> > > > >
> > > > > (2) Does the model know these six values of alpha_V for testing? If the model only sees the training data, it should not know the alpha_V for the test data, which means that alpha_V can be an arbitrary real number. Thus, I am curious about: (i) Does the model perform well for all alpha_V in [-3,3], such as alpha_V = 2.5? If it does, then this is still a 2-D manifold, not a low-dimensional manifold. (ii) Does it work for very large alpha_V such as alpha_V = 10?

---

> > > > > > ### Author Response · Authors · 2022-08-04
> > > > > > **Official Comment**
> > > > > >
> > > > > > We would like to thank the reviewer for your prompt reply and constructive suggestions for the improvement of this paper. It's a pleasure for us to further discuss the experimental details with you.
> > > > > >
> > > > > > ### **Q1 & Q2: KNN's estimation of the source manifold.**
> > > > > >
> > > > > > #### **1. How does kNN work**
> > > > > > **Firstly**,  the k-nearest neighbor graph is a fundamental and basic method to represent the data structure. $K$-nearest neighbors graph is constructed by connecting each sample $x_0$ in the source data with another $K$ samples closest to $x_0$. The resulting KNN graph $G_0$ serves as an approximation for the source manifold by capturing local structures within neighborhoods [1]. As demonstrated in Lemma 1 in [1], it follows that one can approximate geodesic distance from $x_i$ to its neighbors by normalizing distances with respect to the distance to the $k$th nearest neighbor of $x_i$, which shows that kNN is able to capture the manifold. And kNN is also used in many fields to model the data manifold. For example, [2] uses kNN to model the single-cell data and achieves good performance in analyzing the single-cell RNA data.
> > > > > >
> > > > > >
> > > > > > **Secondly**, as for our toy example, in Figure 5 in the revised supplementary material, we showcase the visualized examples of $G_0$, where source data lies on two crossing parabolas. It could be observed that kNN consistently manages to distinguish the two sub-populations until $K=500$. Further, empirical results in the table below prove that with $K<500$ GDRO performs stably better than the baseline with small and moderate $K$, except that smaller $K$ leads to slower convergence since sparse graphs restrain the flow of probability weights. In this experiment, we choose $K=5$ with moderate computing resources.
> > > > > >
> > > > > > | Method       | Test RMSE ($\alpha_V=-0.1$) |
> > > > > > | ------------ | --------------------------- |
> > > > > > | GDRO (K=5)   | 1.666                       |
> > > > > > | GDRO (K=10)  | 1.665                       |
> > > > > > | GDRO (K=20)  | 1.678                       |
> > > > > > | GDRO (K=50)  | 1.756                       |
> > > > > > | GDRO (K=100) | 1.812                       |
> > > > > > | GDRO (K=500) | 1.926                       |
> > > > > > | KL-DRO       | 1.971                       |
> > > > > >
> > > > > > **Thirdly**, manifold learning is an area with intensive research. We have to clarify that manifold learning is not the focus of this paper, and GDRO is compatible with any manifold learning method. We do believe that a more accurate estimated data structure with advanced manifold learning algorithms will further boost the performance of GDRO, and we leave it to future work.
> > > > > >
> > > > > > [1] McInnes, L., Healy, J., & Melville, J. (2018). Umap: Uniform manifold approximation and projection for dimension reduction.
> > > > > >
> > > > > > [2] Dann, E., Henderson, N. C., Teichmann, S. A., Morgan, M. D., & Marioni, J. C. (2022). Differential abundance testing on single-cell data using k-nearest neighbor graphs. Nature Biotechnology, 40(2), 245-253.
> > > > > >
> > > > > >
> > > > > > #### **2. A failure case**
> > > > > >
> > > > > > Here we present an extreme case where KNN achieves poor approximation of the data manifold. When $K$ increases to an extremely large number as $K=500$ in the Toy Example, the neighborhood of KNN diffuses and two manifolds start to merge on the graph, in which case GDRO could not distinguish between two sub-populations and its performance degrades as shown in the table above. Actually in Theorem 3.4 of this paper, we have proved that with an infinitely large $K$ GDRO could be reduced to one of the baseline DROs: KL-DRO, which completely ignores data geometry. Still, we have to clarify that KNN and GDRO perform stably well for a large range of $K$.

---

> > > > > > > ### Author Response · Authors · 2022-08-04
> > > > > > > **Q3: Demonstration of the simulation data.**
> > > > > > >
> > > > > > > Here we address the reviewer's remaining concerns over the two simulation experiments (toy example and selection bias) respectively.
> > > > > > >
> > > > > > > #### **1. Toy Example**
> > > > > > >
> > > > > > > In this experiment, $S$ is the stable feature with a stable relationship with $Y$, and $V$ is the unstable feature whose relationship with $Y$ changes among sub-populations.
> > > > > > >
> > > > > > > **Firstly**, we have to clarify that distributional shifts inside the source manifold is the focus of this paper, and all the other experiments are established in such a setting. Still, to avoid confusion, we supplement the results on the two sub-populations strictly constrained in the manifold of source data ($\alpha_V=1.0$ and $\alpha_V=-0.1$) in the following.
> > > > > > >
> > > > > > > | RMSE         | Major Group ($\alpha_V=1.0$) | Test: Minor Group ($\alpha_V=-0.1$) |
> > > > > > > | ------------ | ---------------------------- | ----------------------------------- |
> > > > > > > | ERM          | 0.992                        | 3.197                               |
> > > > > > > | WDRO         | 0.956                        | 2.563                               |
> > > > > > > | KL-DRO       | 1.374                        | 1.971                               |
> > > > > > > | $\chi^2$-DRO | 1.365                        | 1.981                               |
> > > > > > > | GDRO (K=5)   | 1.655                        | 1.666                               |
> > > > > > >
> > > > > > >
> > > > > > >
> > > > > > > **Secondly**, in the original testing scenarios with $\alpha_V=\{-3,-2,-1,2,3\}$, as for the whole data $X=[S,V]^T$, we admit that the testing data falls out of the estimated manifold. However, we perceive it as a more challenging setting where the data generation mechanism of the spurious variable $V$ changes (this makes the testing data fall out of the source manifold), but the stable one still remains the same as training. This setting is a typical setting in OOD generalization [1]. And from the results, we find that our GDRO could deal with this challenging setting.
> > > > > > > And we will investigate what will happen if the stable variable $S$ also falls out of the manifold in the added experiment below.
> > > > > > >
> > > > > > > **Thirdly**, we would like to demonstrate why GDRO could deal with the setting where $V$ changes a lot (even when it makes the data out of the source manifold). From the results in Table 1 (Est Error), we find that the estimation error of GDRO is quite low, which validates that GDRO could "differentiate" the stable variables from the unstable ones. And the resultant prediction model mainly uses the stable variable $S$ for prediction. Therefore, changing $V$ would not affect much.
> > > > > > >
> > > > > > >
> > > > > > >
> > > > > > > **Further**, we admit that for the simplicity of the experimental setting, we make the stable feature $S$ one-dimensional, which does not form a typical manifold structure. Therefore, we revised the toy example into a 3-D version with a simple extension:
> > > > > > > $$
> > > > > > > S_1 \sim \mathcal{N}(0,1),\quad S_2 = S_1 + \mathcal{N}(0,0.1),\quad Y=\alpha_S(S_1+S_2)+S_1^2+\mathcal{N}(0,0.1),\quad V=\alpha_V Y +\mathcal{N}(0,1).
> > > > > > > $$
> > > > > > > As in the 2-D case, we set $\alpha_V$ in the training set as:
> > > > > > > $$
> > > > > > > \alpha_V=1, \text{with probability }1-r;\quad \alpha_V=-0.1,\text{with probability }r.
> > > > > > > $$
> > > > > > > and $r$ is set to 0.05.
> > > > > > > In this setting, the whole manifold of data $X=[S_1,S_2,V]^T$ with arbitrary $\alpha_V \in \mathbb R$ is approximately the plane $S_1=S_2$. The RMSE of different methods are shown in the following table:
> > > > > > >
> > > > > > > | RMSE         | Major Group ($\alpha_V=1.0$) | Minor Group ($\alpha_V=-0.1$) | $V$ Shift ($\alpha_V=-1.0$) | $S$ Shift ($\alpha_V=-0.1$) |
> > > > > > > | ------------ | :--------------------------: | :---------------------------: | :-------------------------: | :-------: |
> > > > > > > | ERM          |            1.081             |             3.392             |            6.136            |   3.789   |
> > > > > > > | WDRO         |            1.041             |             2.817             |            4.989            |   3.098   |
> > > > > > > | KL-DRO       |            1.441             |             2.114             |            3.280            |   4.036   |
> > > > > > > | $\chi^2$-DRO |            1.409             |             2.196             |            3.536            |   2.450   |
> > > > > > > | GDRO         |            1.709             |             1.762             |            1.832            |   2.386   |
> > > > > > >
> > > > > > > And in this new setting, we could investigate the aforementioned two kinds of 'out-of-manifold' respectively induced by stable variable $S$ and unstable variable $V$:
> > > > > > >
> > > > > > > * The correlation between stable variables shifts so that data is out of the source: we set $S_2=-S_1+\mathcal{N}(0,1)$ (named by "$S$ Shift");
> > > > > > > * The correlation between label and the unstable variable shifts so that data is out of the source: we set $\alpha_V=-1.0$ (named by "$V$ Shift").
> > > > > > >
> > > > > > > From the results, we can see that our GDRO could to some extent deal with the "unstable variable shift" setting, while its performance drops a lot when the stable variables fall out of the estimated manifold.
> > > > > > >
> > > > > > >
> > > > > > > [1] Arjovsky, M., Bottou, L., Gulrajani, I., & Lopez-Paz, D. (2019). Invariant risk minimization.

---

> > > > > > > > ### Author Response · Authors · 2022-08-04
> > > > > > > > **Q3.2 Demonstration of the selection bias experiment**
> > > > > > > >
> > > > > > > > We clarify that in the selection bias experiment, the testing data are constrained in the source manifold in training. We feel sorry for the ambiguity in our formulation, and here we make more demonstrations.
> > > > > > > >
> > > > > > > > Firstly, we characterize the data distribution in a more clear way, which is strictly equivalent to the original selection mechnism ($P(x_i,y_i)=|r|^{-5*|y_i-\text{sign}(r)v_i^b|}$).
> > > > > > > > $$
> > > > > > > > S\sim \mathcal{N}(0,2\mathbb{I}_5),\quad Y=f(S)+\mathcal{N}(0,5),\quad V \sim \text{Laplace}(\text{sign}(r)\cdot Y, \ \ \frac{1}{5\ln |r|})
> > > > > > > > $$
> > > > > > > > Notably that this data generation is exactly equivalent to the original one. And here we can see that $\text{sign}(r)\cdot Y$ is the location parameter, and $\frac{1}{5\ln |r|}$ is the scale parameter. Therefore, in the training data, we let $r_1=1.7$ and $r_2=-1.3$, which makes $V \sim \text{Laplace}(+Y, \ \ \frac{1}{5\ln 1.7})$ in the major group and $V \sim \text{Laplace}(-Y, \ \ \frac{1}{5\ln 1.3})$ in the minor group. And the manifold structure is similar to the original toy example in that the training data is the union of two sub-spaces.
> > > > > > > >
> > > > > > > > Secondly, the sign of $r$ controls the center of $V$, but the main difference with the original toy example lies in the source of distributional shifts. The noise scales of $V$ vary between source and target: the center of $V$ does not change, since it remains $+Y$ or $-Y$. And $|r|$ controls how close $V$ is to the center.
> > > > > > > > As for the testing data where $r=-3.0$, we have $V \sim \text{Laplace}(-Y, \ \ \frac{1}{5\ln 3.0})$, which locates at $-Y$ even more closely. The main difference with the original toy example is that the center of $V$ does not change, since it remains $+Y$ or $-Y$.

---

> ### Comment · Reviewer_aH5S · 2022-08-05
> **Re: New Official Comment**
>
> I want to thank the authors for the new response. I create a new post for this reply so that we don't need to click through all the links to show all the replies. I want to summarize my suggestions for improving this paper as follows:
>
> 1. Based on the new response, as I understand, the authors want to show two things: (i) GDRO does well inside the manifold (like alpha_V = 1 or -0.1 in the first experiment); (ii) GDRO can even do well outside the manifold, i.e. has good OOD generalization (like alpha_V = -3, -2, -1, 2, 3). These are two very different arguments. Argument (i) corresponds to subpopulation shift where the data support does not change. Argument (ii) corresponds to OOD generalization where the data supports of train and test are different. One of the main reasons why the current experiment section is so confusing is that the authors do not distinct between these two, so there seems to be a huge gap between the experiments and the authors' motivation. Therefore, I suggest the authors make a clear distinction in the new version.
>
> 2. It is probably a good idea to completely abandon the first experiment as it is very confusing because: (i) The same as point 1; (ii) It is low-dimensional (even with 3-D). I think the authors can make the Selection Bias as their first experiment which can probably make a better demonstration, but the description of this experiment still needs a lot more improvement.
>
> 3. The description of kNN in the authors' new response is good, so I suggest the authors include them together with the visualization in the main body of the paper, because this is an important component of their method.
>
> 4. As other reviewers have also pointed out, the writing of this paper really needs huge improvement. I am very familiar with DRO, but I still have quite a few difficulties understanding the paper, and people who are not so familiar with DRO could have even more difficulties.
>
> I am happy to further discuss with the authors. Though I still think that the current manuscript needs huge improvement, I am willing to raise my rating if the authors could update the paper as suggested.

---

> > ### Author Response · Authors · 2022-08-07
> > **Official Comment**
> >
> > We would like to thank the reviewer for your prompt reply and constructive suggestions for the improvement of this paper. **We revise our paper according to your suggestions**, especially the experiment section (we highlight the parts revised in the main body).
> > We summarize our revision as:
> >
> > * For  `Section 4. Experiments`:
> >
> >    **(1)** In Section 4, we record implementation details to clarify the settings of experiments, including the construction of KNN from source data only, loss functions for different tasks, hyperparameters during training, and technical designs to support DNN and GPU parallelization.
> >
> >
> >    **(2)** We abandon the toy example as suggested in the 2nd comment so that we focus on inside manifold generalization to clarify the setting (all experiments are the inside manifold setting). Also, half of the experiments are established in a high-dimensional setting ($d \geq 300$).
> >
> >    **(3)** We put the Selection Bias as the first simulated experiment as suggested in the 2nd comment. We **rewrite** the `Data Generation` of the selection bias experiment from the perspective of sub-population shift to improve readability, and we clarify the `Simulation Settings` of this experiment.
> >
> >    **(4)** In Section 4.1, we add one paragraph (`Discussion on kNN`) to discuss the results of kNN in detail, as suggested in the 3rd comment. We showcase the *visualization results* of kNN in the Selection Bias experiment in **Figure 2**, discuss its accuracy and stability w.r.t. $k$, and present an extreme *failure case* of kNN. And we add the results of GDRO under different choices of $k$ in **Table 1**.
> >
> >    **(5)** In Section 4.1, to support the motivation of this paper: data lying in a low-dimensional manifold is embedded in high-dimensional space, we add one high-dimensional ($d=300$) simulated experiment as suggested by the reviewer, where high-dimensional data lie in a low-dimensional manifold. We report the results of all methods in **Table 2**.
> >
> >    **(6)** In Section 4.2, to improve the readability, we add more details and explanations of the problem setting of our real-world experiments (Ionosphere data and Colored MNIST data).
> >
> > * For `Section 3. Proposed Method`:
> >
> >     (1) In Section 3.1, we add one paragraph to clarify `how is $G_0$ estimated` to avoid any ambiguity on this.
> >
> >     (2) In Section 3.1~3.2, we carefully refine our writings to avoid the ambiguity caused by notations.
> >
> > * For `Section 2. Preliminaries`: we extensively discuss more related DRO works and their relationship with GDRO.
> >
> >
> > As for the generalization to data out of the manifold, this work focuses on the inside manifold generalization, which corresponds with our intuitions. The toy example setting assumes some specific invariance structures in it, which is not a general setting to test the out-of-manifold generalization and is low-dimensional. Thus, we agree with the reviewer's suggestion that we *completely abandon the toy example* to avoid confusion, and *replace it with the high-dimensional simulation as suggested*. In this revised main body, **all experiments are the inside manifold setting**. Further, we also discuss the out-of-manifold generalization in Appendix A.8 (due to the space limits).

---

> > > ### Comment · Reviewer_aH5S · 2022-08-07
> > > **Re: Official Comment**
> > >
> > > I would like to thank the authors for the paper revision and the discussions. For the revised version, I would like to raise my rating to accept.

---

> > > > ### Author Response · Authors · 2022-08-08
> > > > **Thanks for your support**
> > > >
> > > > Thank you for your support!
> > > > We appreciate your efforts in all the constructive suggestions and discussions that help to improve this paper.

---

### Official Review · Reviewer_XN8V · 2022-07-10

**Rating:** 8
**Confidence:** 4
**Soundness:** 3 good
**Presentation:** 3 good
**Contribution:** 3 good

**Summary:**

This paper considers the data geometry in the distributionally robust optimization (DRO) problems and proposes a novel framework called Geometric Wasserstein DRO (GDRO) to achieve their goal. The authors also provide some theoretical analyses, such as approximated optimization error and convergence rate, to theoretically show the strengths of GDRO. Finally, the experimental results show the effectiveness of the proposed approach.

**Questions:**

My major concerns are listed below:
- The authors should provide more discussions related to their work, including but not limited to the following concepts:
a)	Distributionally Robust Optimization with Markovian Data.
b)	Orthounimodal Distributionally Robust Optimization.
c)	Distributionally robust shape and topology optimization.
- Can the GDRO apply to large DNNs?
- In Thm.3.1, $\ell(\theta)$ is assumed to be convex. What about the non-convex?

==================================

My concerns about this work have been well addressed. I am happy to vote for its acceptance.

**Strengths And Weaknesses:**

Strengths:
- The motivation and contributions are good. This work attacks an overlooked issue in the DRO community and proposes a reasonable method to alleviate this. This would raise much more research attention in this direction.
- This paper presents some roughly decent theoretical guarantees, which support the effectiveness theoretically.

Weaknesses:
- This paper lacks comprehensive discussions about the related work. As we all know, DRO has attracted tremendous research interest in the machine learning community, and there are many studies about DRO.
- The proposed GDRO may has somewhat limitations, since it is not easy to apply to the deep neural networks (DNNs) (Perhaps I am wrong, but at least the authors do not mention it).

As a whole, despite some limitations, I believe the proposed method is a qualified work and could bring some new insights into the DRO community. Therefore, I tend to accept this paper.

---

> ### Author Response · Authors · 2022-08-02
> **Official Comment**
>
> We sincerely appreciate your approval of the novelty and the theoretical contributions of this work and thank you for the recommendation of the related papers. And we address your concerns in the following:
>
> #### 1. Comprehensive discussions about the related work
> Thanks for recommending us these papers.
> [1] studies the DRO problem for data generated by a time-homogeneous, ergodic finite-state Markov chain and derives the optimization algorithm for it. [2] belongs to the subtype of DRO which constrains the uncertainty set via shape constraints. As for shape constraints, one commonly used is unimodality, and [2] proposes to replace it with the orthounimodality to constitute the uncertainty set for DRO for multivariate extreme event analysis. Compared with the shape constraints, our proposed GDRO intrinsically considers the data geometric properties via Geometric Wasserstein distance, and the geometric property is learned in a data-driven way which is compatible with more complicated machine learning methods, including manifold learning and graph learning methods. Besides, we have added a more thorough discussion of the related works in the revised supplementary material (Appendix A.1).
>
> (ps. sorry that we have not found the third paper named "Distributionally robust shape and topology optimization.")
>
> [1] Distributionally Robust Optimization with Markovian Data.
>
> [2] Orthounimodal Distributionally Robust Optimization.
>
> [3] Distributionally robust shape and topology optimization.
>
>
> #### 2. Apply GDRO to deep neural networks
> * Firstly, GDRO is compatible with any parameterized model including DNN. Compared with ERM, the extra cost of GDRO consists of: 1) The construction of a k-nearest neighbor graph at the initialization step which is built once and for all. 2) The solution of sample weights by gradient flow which is implemented in a way similar to message propagation (since the samples weights are transferred through the edges), which scales *linearly* with sample size and accommodates parallelization by GPU (see implementation details in Appendix B in the revised supplementary material). 3) Model's parameters are updated with a weighted loss by vanilla gradient descent or stochastic gradient descent through batch training, as is usual for DNN.
> * Secondly, we have adopted MLP in our experiments on Colored MNIST, HIV, and Ionosphere data, which empirically demonstrates the adaptability of our model to DNNs.
>
> #### 3. Non-convex loss function
> We have proved that Theorem 3.1 also holds for non-convex loss functions. And we have updated the proof in the revised supplementary material (Appendix A.3).
>
>
> #### 4. Limitations of GDRO
> This work focuses on incorporating data geometric properties into the DRO framework and deriving the corresponding optimization algorithm, and we use the simple k-nearest-neighbor algorithm to learn the graph $G_0$. For more complicated data, the k-nearest-neighbor algorithm may be ineffective and we may need more advanced manifold learning or graph learning methods. Our GDRO framework is compatible with any of these methods, and we leave this for future work.

---

> > ### Comment · Reviewer_XN8V · 2022-08-07
> > **Thank the authors for the detailed clarification**
> >
> > Thank the authors for the detailed clarification. My concerns about this work have been well addressed. I am happy to vote for its acceptance.

---

> > > ### Author Response · Authors · 2022-08-07
> > > **Thanks for your support**
> > >
> > > Thank you for your support! It’s our pleasure to work with you during the rebuttal.

---

### Official Review · Reviewer_wQ3n · 2022-07-11

**Rating:** 7
**Confidence:** 4
**Soundness:** 3 good
**Presentation:** 3 good
**Contribution:** 3 good

**Summary:**

This paper proposed a novel Geometric Wasserstein DRO (GDRO) method by exploiting the discrete Geometric Wasserstein distance. A generically applicable approximate algorithm is derived for model optimization. Extensive experiments on both simulation and real-world datasets demonstrate its effectiveness.

**Questions:**

1. Since the manifold is constructed by the training set, is it still applicable for unseen distributions? Data from unseen distributions may fall out of the data manifold.
2. Does the graph need to be updated at every iteration? If so, it would be time-consuming to estimate the manifold for large-scale datasets.


**Limitations:**

Yes.

**Strengths And Weaknesses:**

Pros:
1. The proposed method is well motivated and reasonable. This paper studied an important problem of DRO: the uncertainty set is too over-flexible such that it may include implausible worst-case distributions. To address this issue, the authors proposed to use Discrete Geometric Wasserstein distance to construct the uncertainty set, in order to constrain the uncertainty set within the data manifold. The method is somewhat novel and interesting.

2. Both convergence rate and the bounded error rate are provided. And the superiority of the proposed method is also empirically demonstrated through experiments on both simulation and real-world datasets.

Cons:
1. Data from unseen distributions may fall out of the manifold constructed by training data. In this case, simply constraining the uncertainty set may not be helpful for OOD generalization.

2. Training efficiency. The authors use a graph to represent the manifold structure. It may be problematic for large-scale datasets since the graph needs to be estimated at every iteration.

3. In the experiments, the authors only compare with ERM and DRO-based methods. It would be a bonus if some general methods for  OOD generalization can be included.

---

> ### Author Response · Authors · 2022-08-02
> **Official Comment**
>
> We sincerely appreciate your approval of both the motivation and novelty of this work and thank you for the advice to compare GDRO with general OOD methods beyond DRO. We would like to address your remaining concerns over this paper.
>
> 1. **Support shift**: GDRO handles unseen distributions with various categories of distributional shifts as stated in the experiment section, including domain shifts, label shifts, and sub-population shifts. However, generalization to target data entirely falling out of the training data's manifold, known as the support shift or non-overlapping support, is not the focus of this work, since it is intrinsically hard to achieve without additional data or assumptions. Firstly, in standard supervised learning, covariate shift with arbitrary support is known to be intractable [1]. Some domain adaptation methods, such as invariant representation learning [2], could empirically validate the effectiveness out of support, but they still require the target distribution to be close to the source's and make strong assumptions about the data. And such methods have also utilized unlabeled target data from unseen support. Secondly, there are some DRO methods allowing for distributions with different support from the original training distribution, such as Wasserstein DRO. However, as mentioned in our paper, extending the support is quite hard in general. For example, in our selection bias simulated experiment, we visualize the results of WDRO in Figure 2 (a), which shows that although WDRO 'creates' some data points, it introduces much more label noises (red points in the third subfigure in Figure 2 (a)). Therefore, this is temporarily not the focus of this work, and we leave it for future work.
>
> 2. **Training efficiency**: Training efficiency is completely not a concern for GDRO. Firstly, the graph is constructed **once and for all** at the initialization step and is fed as input to the algorithm. Secondly, we adopt NN-Descent to construct the k-nearest neighbor graph with an almost linear complexity of $O(n^{1.14})$ for large-scale datasets. Furthermore, since the sample weights are transferred along the edges of the graph, the simulation of gradient flow can be implemented in a way similar to message propagation, which scales linearly with sample size.  The implementation above ensures the adaptability of GDRO to large-scale data. More detailed descriptions of algorithm implementation could be found in the revised supplementary material (Appendix B).
>
> 3. **Compare with other general OOD generalization methods**: Thanks for your advice. Since our GDRO does not utilize additional data or make strong assumptions about the data, we find most of the general OOD generalization methods not suitable for comparing with the proposed GDRO. For example, most of the invariant learning and domain generalization methods require data from multiple environments in training; domain adaptation methods require additional information on the target domain. Therefore, for a fair comparison, we choose the `Environment Inference for Invariant Learning (EIIL)[3]` as another baseline from other branches of OOD generalization, and we test its performance on the Colored MNSIT data. The results are as follows:
>
> | Colored MNIST | Train | Test  |
> |---------------|-------|-------|
> | ERM           | 0.867 | 0.116 |
> | WDRO          | 1.000 | 0.335 |
> | KL-DRO        | 1.000 | 0.287 |
> | $\chi^2$-DRO  | 0.839 | 0.420 |
> | EIIL          | 0.740 | 0.596 |
> | GDRO          | 0.717 | 0.696 |
>
> And we would add this baseline for all experiments in the final version.
>
> [1] David, S. B., Lu, T., Luu, T., & Pál, D. (2010). Impossibility theorems for domain adaptation. In *Proceedings of the Thirteenth International Conference on Artificial Intelligence and Statistics* (pp. 129-136). JMLR Workshop and Conference Proceedings.
>
> [2] Zhao, H., Des Combes, R. T., Zhang, K., & Gordon, G. (2019). On learning invariant representations for domain adaptation. In *International Conference on Machine Learning* (pp. 7523-7532). PMLR.
>
> [3] Creager, E., Jacobsen, J. H., & Zemel, R. (2021). Environment inference for invariant learning. In *International Conference on Machine Learning* (pp. 2189-2200). PMLR.

---

> > ### Comment · Reviewer_wQ3n · 2022-08-07
> > **Response to authors**
> >
> > Thank the authors for their detailed response. All my concerns have been addressed. I confirm my positive feedback on this paper and recommend acceptance.

---

> > > ### Author Response · Authors · 2022-08-08
> > > **Thanks for your support**
> > >
> > > Thank you for your support!
> > > Thanks to your suggestions, several parts have been improved in the rebuttal revision:
> > > * We demonstrate the training efficiency in the `Implementation Details` in Section 4 in the main body.
> > > * We add some discussions on related areas of OOD generalization in Appendix A.1.
> > > * We also discuss the out-of-manifold generalization in Appendix A.8.

---

> > > > ### Comment · Reviewer_wQ3n · 2022-08-08
> > > > **Appreciate the authors' great efforts.**
> > > >
> > > > I would like to raise my score to 7.

---

### Official Review · Reviewer_PrzG · 2022-07-15

**Rating:** 5
**Confidence:** 4
**Soundness:** 2 fair
**Presentation:** 2 fair
**Contribution:** 2 fair

**Summary:**

In this paper, the authors study DRO with data geometry considered in the uncertainty set, making use of the so-called Geometric Wasserstein distance. The authors derive an approximate algorithm for the proposed GDRO and prove its convergence. Numerical experiments are performed to demonstrate the proposed GDRO framework over ERM and other DRO frameworks.

**Questions:**

The mathematics of this article is very hard to follow due to abuse of notation. For example, $p$ has been used to represent a lot of different notions. Various claims are also stated without proof or sufficient explanations (e.g., line 151). The use of English language should also be largely improved. The theoretical results also appear to be very trivial. The description of the experiments must be improved—we don’t even know what loss functions are used in the experiments. The exact DRO problems solved in the experiments, especially for the real-world data ones, are completely unstated. The authors also have to rectify the following issues:

- Typesetting or formatting issues (non-exhaustive):
    - You have to add spaces before all parentheses and citations
    - Add punctuations at the end of display style equations
    - Format tables according to the instructions in the NeurIPS paper template—no vertical lines!
    - Use italics instead of bold to emphasize words

- Typos or writing issues (non-exhaustive):
    - Line 42: $\mathcal{X}_2$—do you mean $\chi^2$?
    - Line 60: $O(1/\sqrt{T})$
    - Line 109: large-scale~~d~~
    - Line 128: ~~cross section~~ cross-sectional
    - Line 129: ~~algorithmic~~ arithmetic
    - Line 137: Brenier
    - Line 146: $\delta(x_i)$ vs $\delta(x_i, y_i)$
    - Line 159: ~~alternate~~ alternating
    - Line 161: ascent~~s~~
    - Line 193: $\mathcal{R}_n$ vs $R_n$
    - Line 200: $\mathcal{GW}$ vs $GW$
    - Line 204: what is $\mathcal{R}(p)$? Not previously defined.
    - Table 1: why adding underscores between “mean” and “error”? Use a dot for abbreviations instead of an underscore.


**Limitations:**

Relevant discussion of limitations might appear in the paper but is hard to find.


**Strengths And Weaknesses:**

Strengths:
- An interesting framework proposed using the Geometric Wasserstein distance
- Extensive experiments

Weaknesses:
- The whole paper is quite hard to follow due to:
    - the lack of self-containedness—some rather non-standard notions are not well-defined or well-explained, or very hard to understand
    - unsatisfactory English writing (wrong grammar, choice of words, etc.)
    - inaccuracy in mathematics exposition, the lack of mathematical rigor and lots of abuse of notation
- Related works are not discussed in detail, e.g., DRO
- A lot of typos, typesetting and formatting issues
- The settings and descriptions of the experiments are unclear


=============================
Post-rebuttal: Thanks the authors for their effort. The revised version has addressed my concerns and I have raised my score. Although it is allowed in this NeurIPS submission cycle, I think the authors should have provided sufficient experimental details in their initial submission. A lot of new content have been added in the revised version of the paper, including supplementary material, say for the experimental details. I feel it is quite unfair to assess the merits of the paper according to this updated version, comparing to other paper submissions. It appears to me that the authors was submitting unfinished work on the submission deadline and take advantage of the rebuttal phase.

---

> ### Author Response · Authors · 2022-08-02
> **Official Comment**
>
> ### 1. Language and formatting issues
> Thanks for your suggestions on the language and formatting issues in this paper. Most typos are immediately fixed in the rebuttal revision. While a few of them remain unchanged for the following considerations:
>
> - Line 146: The empirical *marginal* distribution of $X$ is formulated as $\sum_{i=1}^n\delta(x_i)$ instead of  $\sum_{i=1}^n\delta(x_i, y_i)$.
> - Line 204: $\mathcal R(p)$ is stated as the overall *objective function* of this paper, and its definition can be traced back to Equation 4. We have claimed $\theta$ and $n$ as constants in the context,  and to avoid any ambiguity, we explicitly specify $\mathcal R(p)$ as the abbreviation for $\mathcal R_n(\theta, p)$ in the rebuttal revision.
>
> Additionally, we perform an exhaustive re-examination of the paper to ensure that the revision is free of similar typos. Apart from the suggestions on language or style, we would really appreciate if you could further offer detailed comments on the *technical* content.
>
> ### 2. Explanation of notations and claims
> Regarding the obstacles you encountered while going through the paper, we provide some detailed explanations below:
>
> - The notation $p$ is consistent through the article, representing a continuously differentiable curve $p(t) : [0, 1] → \mathscr P_0 (G_0 )$ which describes the transformation of a measure on the vertex space of graph $G_0$ (see Definition 3.1). And the probability weight of the $i$-th vertex at time $t$ is abbreviated as $p_i(t)$ (see Line 165, clarification of notations). When $t$ is clear from the context, we denote the probability weight of the $i$-th vertex by $p_i$. We discuss WDRO in section 2 (related work), which considers the transformation of a measure in the Euclidean space instead of the discrete space of GDRO. In this case, $p(t) : [0, 1] → \mathscr P_0 (\mathbb R^n)$ naturally induces the transport plan $p(t,x):([0,1],\mathbb R^n) → \mathbb R$ which is the probability density of the point $x$ at time $t$ (see Line 92). The notations are common in optimal transportation theory based on which GDRO is developed (see reference [21]: *Topics in optimal transportation*).
>
> - The claim in Line 151 that the objective of GWRO (Equation 4) degenerates to a $f$-DRO as $\beta → \infty$ is straightforward from the fact that both $f$-DRO and GWDRO are equivalent to ERM with infinitely large $\beta$. We've included the simple proof in the revised supplementary material.
>
> ### 3. Details of experiments
> - The OOD setting is emphasized for each experiment. Further implementation details are elaborated in the supplementary material. Below is a table from the Appendix summarizing the various distributional shifts that Distributionally Robust Optimization shall handle for each experiment. We take the real-world datasets for example: the Retiring Adults dataset is targeted at subpopulation shift; Colored MNIST is targeted at concept shift, and the IonoSphere and HIV datasets are targeted at label shift. We comprehensively discuss the sources, intensity, and mechanics of each distributional shift for each experiment in section 4. In terms of loss functions, we adopt MSE for the regression task and cross-entropy for the classification task. We omit the specification of the trivial ERM objective in the original paper. Further implementation details are elaborated in the revised supplementary material, which will be added to the main body when accepted.
>
>
> | Data          |  Toy Example | Selection Bias | Colored MNIST | Retiring Adults |     HIV    | Ionosphere | Added Simulation Data |
> |---------------|:------------:|:--------------:|:-------------:|:---------------:|:----------:|:----------:|:---------------------:|
> | Kind          |  Simulation  |   Simulation   |      Real     |       Real      |    Real    |    Real    |       Simulation      |
> | Dimension     |       2      |       10       |      2352     |      10~19      |     160    |     34     |          300          |
> | Shift Pattern | Sub-population |  Domain Shift  |  Domain Shift |  Sub-population | Label Shift | Label Shift |      Domain Shift     |
> | Model         |    Linear    |     Linear     |      MLP      |      Linear     |     MLP    |     MLP    |         Linear        |
>
>
>
> ### 4. Related works
> In terms of related work, Section 2 reviews the literature on DRO. Since DRO methods are characterized by the uncertainty set they adopt for robust optimization. We classify the most relevant DRO literature into two categories based on the distance metric to specify the uncertainty set: $f$-divergence and Wasserstein distance. More detailed discussions on extensive DRO methods and other general OOD works are included in the revised supplementary material.

---

> > ### Author Response · Authors · 2022-08-02
> > **Official Comment**
> >
> > ### 5. The novelty of this paper
> > We disagree with the comments that theoretical results appear to be very trivial.
> >
> > The novelty of this paper includes: (1) We firstly incorporate the data geometric properties into the DRO framework in a data-driven way to address the over-pessimism problem in DRO. Our method is data-driven and friendly to deep neural networks. This idea has hardly been explored, yet can largely overcome the over-pessimism problem. Further, it opens up a promising new avenue for DRO to naturally incorporate manifold learning and graph learning methods. (2) We derive the approximate optimization algorithm for the newly-proposed objective and characterize its error rate as well as the convergence rate. All the other reviewers acknowledged our contributions, especially the reviewer aH5S who perceived that *a lot of credit should be given to the authors for the theory part: nicely formulated GDRO objective and convergence result.*
> >
> > At last, we welcome any **technical** advice for our proposed method and insights into the underlying theory. And we're ready to address your concern if you still find the paper hard to follow.

---

### Meta-Review · Area_Chair_ivvK · 2022-08-26

**Recommendation:** Accept
**Confidence:** Certain

**Metareview:**

The paper proposes a novel distributionally robust optimization formulation leveraging data geometry to construct the uncertainty set. After a lengthy discussion and revision process, the reviewers have reached a consensus acceptance recommendation, which I support.

Currently the reproducibility checklist (part 3a) states that the authors submitted code to reproduce their results along with the paper, but I do not see it as part of the supplementary material or as a link. Please provide code with the camera ready submission, or correct the checklist.

**Award:**

No

---

### Decision · Program_Chairs · 2022-09-14

Accept